# Benchmarking the Spectrum of Agent Capabilities

**Danijar Hafner**
Google Research, Brain Team
University of Toronto
`mail@danijar.com`

## Abstract

Evaluating the general abilities of intelligent agents requires complex simulation environments. Existing benchmarks typically evaluate only one narrow task per environment, requiring researchers to perform expensive training runs on many different environments. We introduce Crafter, an open world survival game with visual inputs that evaluates a wide range of general abilities within a single environment. Agents either learn from the provided reward signal or through intrinsic objectives and are evaluated by semantically meaningful achievements that can be unlocked during each episode, such as discovering resources and crafting tools. Consistently unlocking all achievements requires strong generalization, deep exploration, and long-term reasoning. We experimentally verify that Crafter is of appropriate difficulty to drive future research and provide baselines scores of reward agents and unsupervised agents. Furthermore, we observe sophisticated behaviors emerging from maximizing the reward signal, such as building tunnel systems, bridges, houses, and plantations. We hope that Crafter will accelerate research progress by quickly evaluating a wide spectrum of abilities.

## 1 Introduction

Crafter is an open world survival game for reinforcement learning research. Shown in Figure 1, Crafter features randomly generated 2D worlds with forests, lakes, mountains, and caves. The player needs to forage for food and water, find shelter to sleep, defend against monsters, collect materials, and build tools. The game mechanics are inspired by the popular game Minecraft and were simplified and optimized for research productivity. Crafter aims to be a fruitful benchmark for reinforcement learning by focusing on the following design goals:

**Research challenges**   Crafter poses substantial challenges to current methods. Procedural generation requires strong generalization, the technology tree evaluates wide and deep exploration, image observations calls for representation learning, repeated subtasks and sparse rewards evaluate long-term reasoning and credit assignment.

**Meaningful evaluation**   Agents are evaluated by a range of achievements that can be unlocked in each episode. The achievements correspond to meaningful milestones in behavior, offering insights into ability spectrum of both reward agents and unsupervised agents.

**Iteration speed**   Crafter evaluates many agent abilities within a single environment, vastly reducing the computational requirements over benchmarks suites that require training on many separate environments from scratch, while making it more likely that the measured performance is representative of new domains.

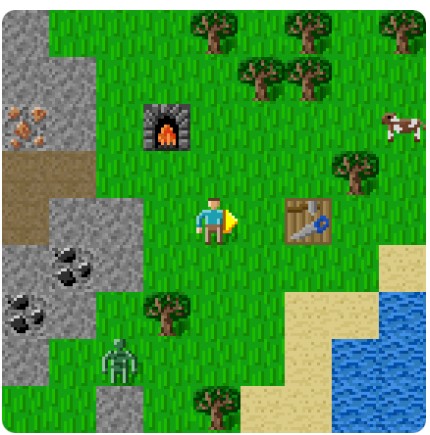

**Figure 1:** Agent view of a procedurally generated world in Crafter, showing terrain types, resources, and creatures. Agents learn from image inputs and aim to unlock a range of semantically meaningful achievements during each episode. The achievements evaluate strong generalization, wide and deep exploration, and long-term reasoning.

```
$ python3 -m pip install crafter    # Install Crafter
$ python3 -m pip install pygame     # Needed for human interface
$ python3 -m crafter.run_gui        # Start the game
```

**Figure 2:** Play Crafter yourself through the human interface.

## 2    RELATED WORK

Benchmarks have been a driving force behind the progress and successes of reinforcement learning as a field (Bellemare et al., 2013; Brockman et al., 2016; Kempka et al., 2016; Beattie et al., 2016; Tassa et al., 2018; Juliani et al., 2018). Benchmarks often require a large amount of computational resources and yet only test a small fraction of the abilities that a general agent should master (Cobbe et al., 2020). This section directly compares Crafter to four particularly related benchmarks.

**Minecraft**    Crafter is inspired by the successful 3D video game Minecraft, which is available to researchers via Malmo (Johnson et al., 2016) and MineRL (Guss et al., 2019). Minecraft features diverse open worlds with randomly generated and modifiable terrain, as well as many different resources, tools, and monsters. However, Minecraft is too complex to be solved by current methods (Milani et al., 2020), it is unclear by what metric agents should be evaluated by, the environment is slow, and can be difficult to use because it requires Java and a window server. In comparison, Crafter captures many principles of Minecraft in a simple and fast environment, where results can be obtained in a matter of hours, and where a large number of semantically meaningful evaluation metrics are available for reinforcement learning with or without extrinsic reward. The goal of Crafter is not to replace Minecraft but progress faster towards it.

**Atari**    The Atari Learning Environment (Bellemare et al., 2013) has been the gold standard benchmark in reinforcement learning. It comprises around 54 individual games, depending on the evaluation protocol (Mnih et al., 2015; Schulman et al., 2017; Badia et al., 2020; Hafner et al., 2020). While the large number of games tests different abilities of agents, they require a large amount of computation. The recommended protocol of training the agent with 5 random seeds on each game for 200M steps requires over 2000 GPU days (Castro et al., 2018; Hessel et al., 2018). This substantially slows down experimentation and makes the complete benchmark infeasible for most academic labs. Moreover, Atari games are nearly deterministic, so agents can approximately memorize their action sequences and are not required to generalize to new situations (Machado et al., 2018).

**ProcGen**    ProcGen (Cobbe et al., 2020) provides a benchmark that is similar to Atari but explicitly addresses the determinism present in Atari through the use of procedural generation and randomized textures. It consists of 16 games, where each episode features a randomly generated level layout. Similarly, Crafter relies on procedural generation to provide a different world map with different distribution of resources and monsters for every episode. However, ProcGen still requires training methods on 16 individual games for 200M environment steps, which each focus on a narrow aspect of an agent's general abilities. In comparison, Crafter evaluates many different abilities of an agent by training only on a single environment for 5M steps, substantially accelerating experimentation.

**NetHack**    NetHack (Küttler et al., 2020) is a text-based game, where the player traverses a randomly generated system of dungeons with many different items and creatures. Unlike the other discussed environments, NetHack uses symbolic inputs and thus does not evaluate an agent's ability to learn representations of high-dimensional inputs. The game is challenging due to the large amount of knowledge required about the many different items and their effects, even for human players. As a result, NetHack requires many environment steps for agents to acquire this domain-specific knowledge; 1B steps were used in the original paper. In contrast, Crafter generates diverse complex worlds from simple underlying rules, focusing more on generalization than memorization of facts.

## 3    CRAFTER BENCHMARK

We introduce Crafter, a benchmark that evaluates a variety of agent abilities in a single environment. This section describes the game mechanics of the environment, the interface of agent inputs and actions, the evaluation protocol that is based on a range of semantically meaningful achievements, and the open challenges that Crafter poses for future research.

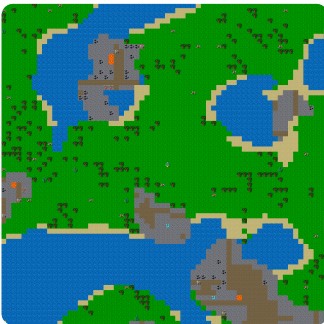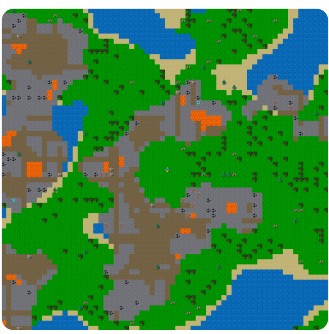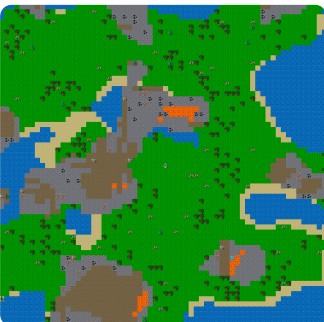

**Figure 3:** Crafter procedurally generates a unique world for every episode that features several terrain types: grasslands, forests, lakes, mountains, caves. Memorizing action sequences is thus not a viable strategy and agents are forced to learn behaviors that generalize to new situations.

## 3.1 GAME MECHANICS

This section describes the game mechanics of Crafter, namely its randomly generated world maps, the levels of health and other internal quantities that the player has to maintain, the resources it can collect and objects and tools it can make from them, as well as the creatures and how they are influenced by the time of day. The images of all materials and objects are shown in Figure E.1. All randomness in the environment is uniquely determined by an integer seed that is derived from the initial seed passed to the environment and the episode number.

**Terrain generation**   A unique world is generated for every episode, shown in Figure 3. The world leverages an underlying grid of $64 \times 64$ cells but the agent only observes the world through pixel images. The terrain features grasslands, lakes, and mountains. Lakes can have shores, grasslands can have forests, and mountains can have caves, ores, and lava. These are determined by OpenSimplex noise (Spencer, 2014), a form of locally smooth noise. Within the areas determined by noise, objects appear with equal probability at any location, such as trees in forests and skeletons in caves.

**Health and survival**   The player has levels of health, food, water, and rest that it must prevent from reaching zero. The levels for food, water, and rest decrease over time and are restored by drinking from a lake, chasing cows or growing fruits to eat, and sleeping in places where monsters cannot attack. Once one of the three levels reaches zero, the player starts losing health points. It can also lose health points when attacked by monsters. When the health points reach zero, the player dies. Health points regenerate over time when the player is not hungry, thirsty, or sleepy.

**Resources and crafting**   There are many resources, such as saplings, wood, stone, coal, iron, and diamonds, the player can collect in its inventory and use to build tools and place objects in the world. Many of the resources require tools that the place must first build from more basic resources, leading to a technology tree with several levels. Standing nearby a table enables the player to craft wood pickaxes and swords, as well as stone pickaxes and stone swords. Crafting a furnace from stone enables crafting iron pickaxes and iron swords from both iron, coal, and wood.

**Creatures and night**   Creatures are initialized in random locations and move randomly. Zombies and cows live in grasslands and are automatically spawned and despawned to ensure a given amount of creatures. At night, the agent's view is restricted and noisy and a larger number of zombies is spawned. This makes it difficult to survive without securing a shelter, such as a cave. Skeletons live in caves and try to keep the player at a distance to shoot arrows at the player. The player can interact with creatures to decrease their health points. Cows move randomly and offer a food source.

## 3.2 ENVIRONMENT INTERFACE

This section defines the specification of the environment, explains the available actions, agent inputs, episode termination, and additional information provided by the environment. The design goal of these is to make the environment easy to use and inspect. The environment uses the Gym interface (Brockman et al., 2016) with visual agent inputs and flat categorical actions.

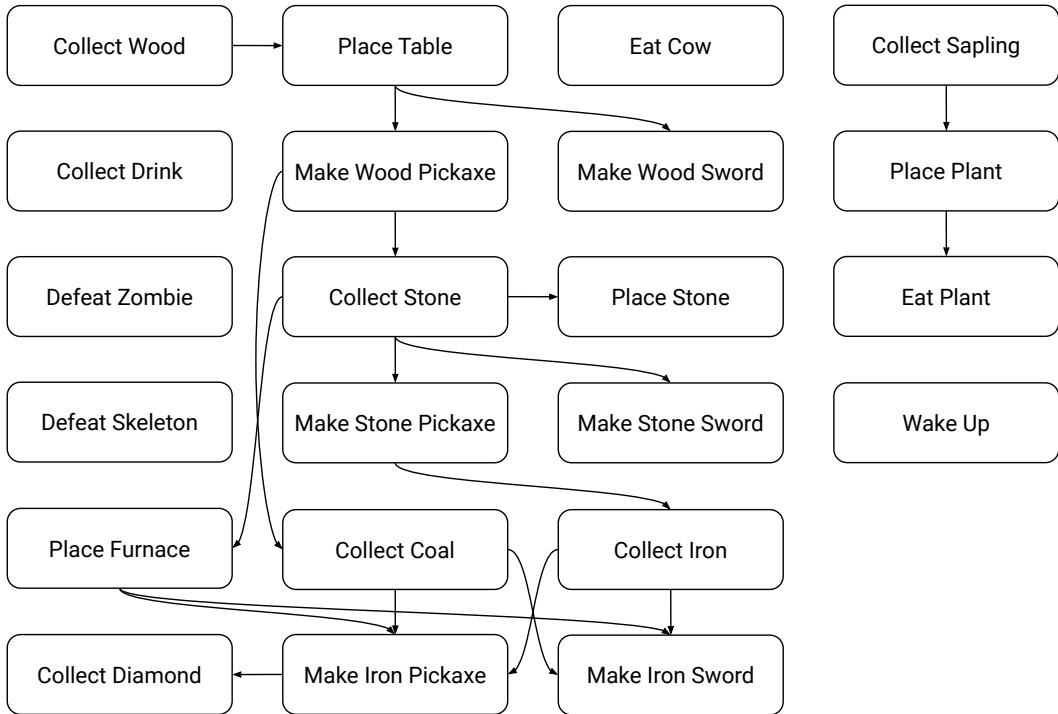

**Figure 4:** The 22 achievements that can be unlocked within each episode. The arrows indicate which achievements will be completed along the way of working toward more challenging achievements. Several of the earlier tasks have to be repeated multiple times, such as collecting resources, to progress further. A reward is only given when an achievement is unlocked for the first time during the episode.

**Observations**  Agent receive color images of size $64 \times 64 \times 3$ as their only inputs. The image shows a local top-down view of the map, reaching 4 cells west and east and 3 cells north and south of the player position. Below this view of the world, the image shows the current inventory state of the player, including its health points, food, water, and rest levels, collected materials, and crafted tools. The agent needs to learn to read its inventory state out of the image.

**Actions**  The action space is a flat categorical space with 17 actions, represented by integer indices. The actions allow the player to move in all 4 directions along the grid, interact with the object in front of it, go to sleep, place objects, and make tools. Each object and tool has a separate action associated with it. Tools are kept in the inventory whereas objects are automatically placed in front of the player. If the agent does not hold the required materials for making an object or tool, the action has no effect.

**Termination**  Each episode terminates when the player's health points reach 0. This can happen when the player dies out of hunger, thirst, or tiredness, when attacked by a zombie or skeleton, or when falling into lava. Health points automatically regenerate, as long as the agent is not too hungry, thirsty, or sleepy. There is no negative reward for dying, as the reward signal already includes a penalty for losing health points. Episodes also end when reaching the time limit of 10,000 steps.

**Additional information**  The environment allows access to privileged information about the world state that the agent is forbidden to observe. This includes numeric inventory counts, achievement counts, the current coordinate of the player on the grid, and a semantic grid representation of the map. These can be used for debugging purposes or for other research scenarios, such as predicting the underlying environment state to evaluate representation learning or video prediction models.

## 3.3 EVALUATION PROTOCOL

To evaluate the diverse abilities of artificial agents on Crafter, we define two benchmarks. The first benchmark allows agents to access a provided reward signal, while the second benchmark does not and requires agents to purely learn from intrinsic objectives. Besides access to the provided reward signal, the evaluation protocols are identical. An agent is granted a budget of 1M environment steps to interact with the environment. The agent performance is evaluated through *success rates* of the individual achievements throughout its training, as well as an aggregated *score*.

**Achievements**   To evaluate a wide spectrum of agent abilities, Crafter defines 22 achievements. The achievements are shown in Figure 4 and correspond to semantically meaningful behaviors, such as collecting various resources, building objects and tools, finding food and water, defeating monsters, and waking up safely after sleeping. The achievements cover a wide range of difficulties, making them suitable to evaluate both weak and strong players and providing continuous feedback throughout the development process of new methods. Some achievements are independent of each other to test for breadth of exploration, while others depend on each other to test for deep exploration.

**Reward**   Crafter provides a sparse reward signal that is the sum of two components. The main component is a reward of $+1$ every time the agent unlocks each achievement for the first time during the current episode. The second component is a reward of $-0.1$ for every health point lost and a reward of $+0.1$ for every health point that is regenerated. Because the maximum number of health points is 9, the second reward component only affects the first decimal of the episode return, and ceiling the episode return yields the number of achievements unlocked during the episode.

**Success rates**   The success rates offer insights into the breadth of abilities learned by an agent. The success rates are computed separately for each of the achievements, as the fraction of training episodes during which the agent has unlocked the achievement at least once. It is computed across all episodes that lead up to the budget of 1M environment steps, requiring agents to be data-efficient.[1] Note that the number of environment steps is fixed but the number of episodes can differ between agents. Unlocking an achievement more than once per episode does not affect the success rate.

**Score**   The score summarizes the agent abilities into a single number. It is computed by aggregating the success rates for the individual achievements. Unlocking difficult achievements, even if it happens rarely, should contribute more than increasing the success rate of achievements that are already unlocked frequently even further. To account for the range of difficulties of the achievements, we average the success rates in log-space, known as the geometric mean.[2] Unlike the reward, the score thus takes the achievement's difficulties into account, without having to know them beforehand.

**Discussion**   Aggregating across tasks via a geometric mean weighs tasks based on their difficulty to the agent, resulting in higher scores for agents that explore more broadly. For example, collecting a diamond 1% of the time instead of 0% is a meaningful improvement, whereas collecting wood 95% of the time instead of 90% is not. This allows distinguishing how broadly agents have explored their environment even if they achieve similar rewards. The geometric mean also establishes a meaningful metric for unsupervised agents, which may get bored of tasks after performing them a few times and then move on to new tasks. A caveat of the geometric mean is that agents with rewards are evaluated by something they only indirectly optimize for, which can change their ranking order. Increasing reward and score is generally correlated, but capacity-limited agents may choose to optimize reward by mastering easy tasks and ignoring hard tasks, which only slowly increases the geometric mean.

### 3.4   RESEARCH CHALLENGES

Crafter aims to evaluate a diverse range of agent abilities within a single environment. Thus, if a method performs well on Crafter there should be a high chance that it also handles the challenges of other environments. The challenges also make Crafter suitable for evaluating progress on open research questions, such as strong generalization, wide and deep exploration, discovering reusable skills, and long-term memory and reasoning. Crafter is designed to pose the following challenges:

**Exploration**   Independent achievements evaluate wide exploration, without offering a linear path for the agent to follow. Dependent achievements evaluate deep exploration of the technology tree. Collecting a diamond requires an iron pickaxe, which in turn requires a furnace, table, coal, iron, and wood. The furnace requires collecting stone, which requires building a wood pickaxe at a table.

**Generalization**   Every episode is situated in a unique world that is procedurally generated. Moreover, many aspects of the game reoccur in different contexts, such as creatures and resources that can be found in different landscapes and times of day. This forces successful agents to recognize similar situations in different circumstances and be robust to changes in irrelevant details.

---

[1]While allowing a large number of environment steps would help agents achieve higher scores more easily, it would result in a comparison of compute resources rather than algorithm quality.

[2]The Crafter score of an agent is computed as $S \doteq \exp(\frac{1}{N}\sum_{i=1}^{N}\ln(1+s_i))-1$, where $s_i \in [0;100]$ is the agent's success rate of achievement $i$ and $N=22$ is the number of achievements.

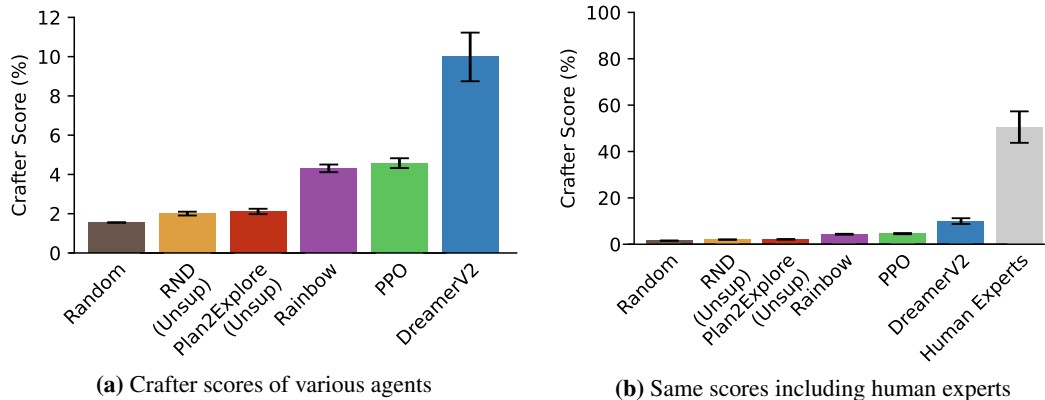

**(a)** Crafter scores of various agents  **(b)** Same scores including human experts

**Figure 5:** Crafter Benchmark Scores for various agents with and without rewards. Current top methods achieve scores of up to 10% that are far from the 50% of human experts, posing a substantial challenge for future research. Crafter scores are computed as the geometric mean across achievements of their success rates within the budget of 1M environment steps. Numbers in Table 1.

**Reusable skills** Advancing in the game requires the agent to repeat several behaviors over long horizons, such as finding food, defending against monsters, and collecting common materials that are needed many times. The behavior of a successful agent naturally decomposes into sub-tasks, making Crafter suitable for studying hierarchical reinforcement learning.

**Credit assignment** Only sparse rewards are given for unlocking an achievement for the first time during each episode. Moreover, several achievements require long-term reasoning, such as collecting the necessary resources for crafting a particular tool or planting saplings that can be harvested many hundred time steps later. This makes Crafter a challenge for temporal credit assignment.

**Memory** The agent inputs only show the player's immediate surroundings, making Crafter partially observed. To survive for a long time, agents need to remember where to find lakes to drink and open grasslands to hunt. Moreover, to effectively find rare resources, such as iron and diamonds, the agent needs to remember what parts of the map it has already searched.

**Representation** The agent observes its environment via high-dimensional images, from which it has to extract entities that are meaningful for decision making. Similar to applications in the real world, the reward signal is sparse and the amount of environment interaction limited. As a result, successful agents will likely rely on explicit representation learning techniques.

**Survival** In previous environments, the player can often survive by doing nothing. This allows for degenerate solutions to intrinsic objectives, unlike the real world where animals are forced to adapt to survive and maintain homeostasis and allostasis. In Crafter, the player struggles to survive through the constant pressure of maintaining enough water, food, rest, and defending against zombies.

| Method | Score (%) | Return |
|---|---|---|
| Human Experts | $50.5\pm6.8$ | $14.3\pm2.3$ |
| DreamerV2 | $10.0\pm1.2$ | $9.0\pm1.7$ |
| PPO | $4.6\pm0.3$ | $4.2\pm1.2$ |
| Rainbow | $4.3\pm0.2$ | $5.0\pm1.3$ |
| Plan2Explore (Unsup) | $2.1\pm0.1$ | $2.1\pm1.5$ |
| RND (Unsup) | $2.0\pm0.1$ | $0.7\pm1.3$ |
| Random | $1.6\pm0.0$ | $2.1\pm1.3$ |

**Table 1:** Crafter benchmark scores. The Crafter score is computed as the geometric mean of success rates for all 22 achievements available in the environment. The score prefers general agents that unlock a wide range of achievements over those that unlock a small number of achievements very frequently. For example, an agent that explores many different achievements over the course of training achieves a higher score than one that only performs same simple tasks over an over. The score thus establishes a meaningful metric both for agents with and without reward.

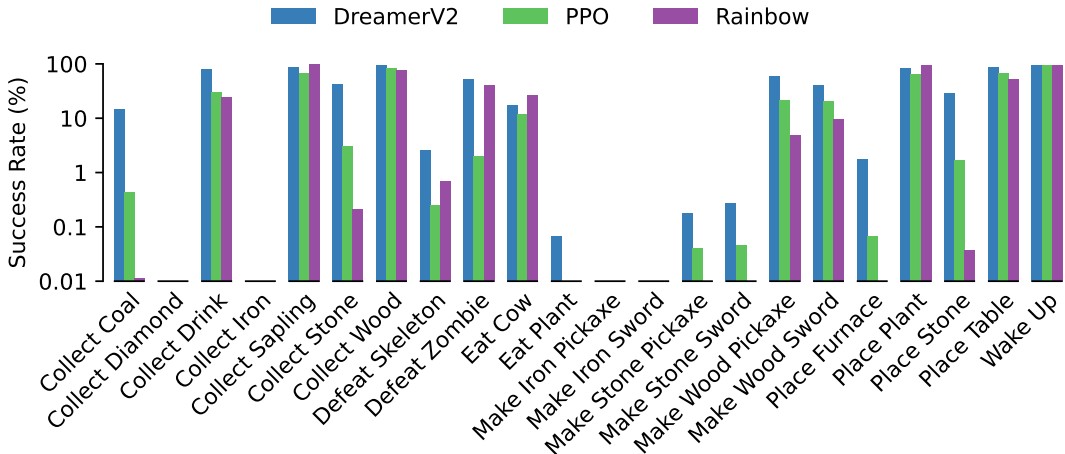

**Figure 6:** Agent ability spectrum showing the success rates of agents with rewards. These are unlocking percentages for all 22 achievements, computed over all training episodes. Rainbow manages to drink water and forage for food. PPO additionally rarely collects coal and builds stone tools. DreamerV2 achieves these more frequently and additionally sometimes grows and eats fruits. Numbers in Appendix A.

## 4 EXPERIMENTS

To established baselines for future work, we train various reinforcement learning methods on Crafter either with and without rewards. The two benchmarks follow the evaluation protocol in Section 3.3, which grants each agent a budget of 1M environment frames and computes the *success rates* of the individual achievements across all training episodes, as well as an aggregate *score* for the agent. Furthermore, we analyze the emergent agent behaviors qualitatively and record a dataset of human expert players to estimate the difficulty of the environment. The environment, code for the baseline agents and figures in this paper, and the human dataset are available on the project website. [3]

### 4.1 BENCHMARK WITH REWARDS

We provide baselines scores for three reinforcement learning algorithms on Crafter with rewards. DreamerV2 (Hafner et al., 2020) learns a world model and optimizes a policy through planning in latent space. We used its default hyper parameters for Atari and increased the model size. PPO (Schulman et al., 2017) is a popular method that learns to map input images to actions through policy gradients. We use a convolutional neural network policy with hyper parameters that were tuned for Atari (Hill et al., 2018). Rainbow (Hessel et al., 2018) is based on Q-Learning and combines several advances, including for exploration. The defaults for Atari did not work well, so we tuned the hyper parameters for Crafter and found a compromise between Atari defaults and the data-efficient version of the method (van Hasselt et al., 2019) to be ideal. All agents trained for 1M environment steps in under 24 hours on a single GPU and we repeated the training for 10 random seeds per method. The training reward curves are included in Appendix D.

The scores are listed in Table 1 and visualized in Figure 5. DreamerV2 achieves a score of 10.0%, followed by PPO with 4.6% and Rainbow of 4.3%. Despite these being top reinforcement learning methods, they lack behind the score of expert human players of 50.5%, which we describe in further detail in Section 4.3. We conclude that Crafter is a challenging benchmark, where current methods make learning progress but future research is needed to achieve high performance. For comparison, we report the episode returns in Table 1, computed over the episodes within the last $10^5$ environment steps of training. We find a trend similar to the scores but notice that the methods are harder to tell apart, because differences on hard tasks that are rarely achieved affect the return less. Moreover, the scores are more meaningful for unsupervised agents, which should explore many achievements over time, but not necessarily remain interested in them until the end of training. The success rates for individual achievements are visualized in Figure 7, which offer insights into the breadth and depth of agent abilities. Rainbow displays high success rates on easier achievements. PPO learned

---

[3] https://danijar.com/crafter

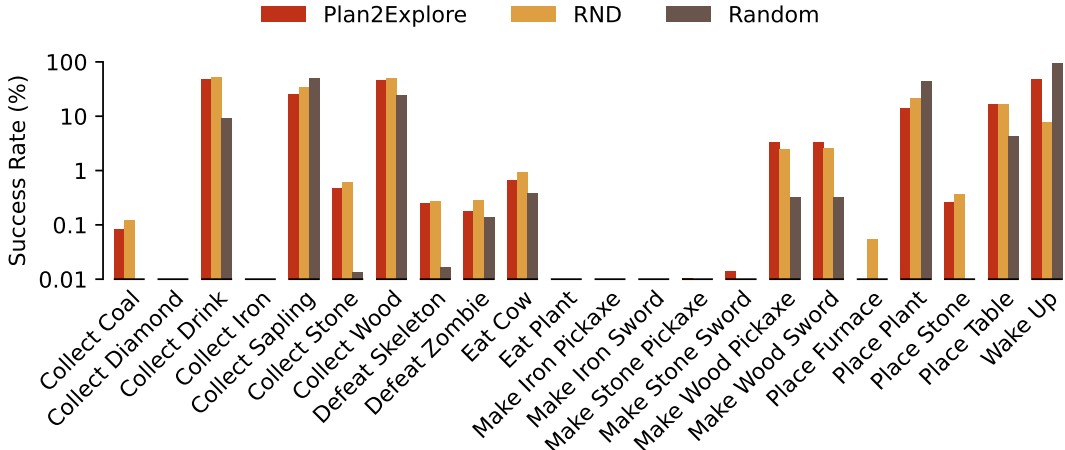

**Figure 7:** Agent ability spectrum showing the success rates for Crafter without rewards. Random actions unlock the 6 easiest achievements sometimes, such as drinking water and collecting wood. Plan2Explore forages for food and defeats monsters more frequently, to ensure longer survival. RND additionally collects stones and rarely even collects coal and builds furnaces. Numbers in Appendix B.

to additionally make stone tools and furnaces. DreamerV2 achieved these more frequently and discovered growing and harvesting plants. None of the agents learned to collect and use iron for tools or to collect diamonds, or to achieve high success rates on many of the achievements.

## 4.2 UNSUPERVISED BENCHMARK

We provide baselines scores for two unsupervised reinforcement learning agents on Crafter without rewards. We also include a baseline that simply chooses random actions. RND (Burda et al., 2018b) is a popular exploration method that seeks out novel inputs, estimated as the prediction error of a network that aims to predict fixed random embeddings of the input images. We use its default parameters for Atari. Plan2Explore (Sekar et al., 2020) learns a world model to plan for the expected information gain of imagined trajectories, allowing it to directly seek out imagined states that have not been experienced before. We implement Plan2Explore on top of DreamerV2 and keep the same hyper parameters. We use a non-episodic value function as RND does, which helps exploration in episodic environments (Burda et al., 2018b). All agents trained for 1M environment steps in under 24 hours on a single GPU and we repeated the training for 10 random seeds per method.

The scores are listed in Table 1 and in Figure 5. Plan2Explore achieves a score of 2.1%, followed by RND at 2.0%, both ahead of the random agent at 1.6%. Despite these being top unsupervised reinforcement learning methods, they lack far behind optimal performance or even the performance of agents that learn with rewards, posing a substantial challenge for future research. The results are encouraging, showing that unsupervised objectives by themselves can lead to meaningful behaviors (Burda et al., 2018a) in Crafter. Inspecting the success rates for individual achievements in Figure 6 confirms that Plan2Explore and RND make progress in exploring the different behaviors compared to the random agent, including occasionally collecting coal, placing furnaces, and making stone swords, which are several steps deep into the technology tree.

## 4.3 EMERGENT BEHAVIORS

To better understand the potential of the environment, we train DreamerV2 for 50M steps and investigate the behaviors qualitatively. In this amount of time, the agent learns to build stone tools and even iron tools on individual occurrences. Interestingly, we observe a range of sophisticated emergent behaviors, such as building tunnel systems, building bridges to cross lakes, and outsmarting skeletons by dodging arrows, blocking arrows with stones, and digging through walls to surprise skeletons from the side. Furthermore, DreamerV2 learns to seek shelter to protect itself from the zombies at night by hiding in caves and even digging its own caves and closing the entrances with stones. Finally, we find that the agent sometimes manages to build plantations of many saplings, defends them against monsters, and eats the growing fruits in order to ensure a reliable and steady food supply. A video of the emergent behaviors is available on the project website.

### 4.4 HUMAN EXPERTS DATASET

Crafter includes a graphical user interface that allows humans to play the game via the keyboard and record the trajectories of the game. The human interface can be installed via the command shown in Figure 2. Through the human interface, we recorded the games of 5 human experts for a combined total of 100 episodes. The experts were given the instructions of the game and allowed several hours of practice. Out of the 100 episodes, 5 episodes unlock all 22 achievements. The human experts achieved a score of 50.5%, unlocking all achievements as shown in Table C.1. The achievements most difficult to humans were to collect diamonds and grow and harvest plans, with success rates of 12% and 8%, respectively. While the human dataset is separate from the Crafter benchmark, it provides an estimate of human performance and can be used for research on learning from demonstrations and imitation learning. The human dataset is available on the project website.

## 5 DISCUSSION

**Future work**   We selected the difficulty of Crafter to be challenging yet not hopeless for current methods. As research progresses towards solving the challenges that are currently present, it may become necessary to extend Crafter by new enemies, resources, items, and achievements. Being written purely in Python, Crafter can easily be extended in this way. Moreover, grouping the 22 achievements into categories, such as *memory*, *generalization*, and *exploration*, would allow us to summarize agent abilities more abstractly (Osband et al., 2019). We did not attempt such a categorization because it is subjective and will become clearer as more researchers use the environment.

**Summary**   We introduced Crafter, a benchmark with visual inputs that evaluates a variety of general agent abilities in a single environment. We described the game mechanics, evaluation protocol, and open challenges posed by the benchmark, and performed experiments with several agents with and without rewards to provide baseline scores. Agents are evaluated based on how frequently they manage to unlock achievements that correspond to semantically meaningful milestones of behavior. We conclude that Crafter is well suited and of appropriate difficulty to guide future research on intelligent agents, both for learning from extrinsic rewards and purely from intrinsic objectives.

**Acknowledgements**   We would like to thank Oleh Rybkin, Ben Eysenbach, Sherjil Ozair, Julius Kunze, Feryal Behbahani, Timothy Lillicrap, Jimmy Ba, Nicolas Heess, Kory Mathewson, Mohammad Norouzi, Hamza Merzic, and Sergey Levine for discussions and feedback.

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

# A  SUCCESS RATES WITH REWARDS

| Achievement | Rainbow | PPO | DreamerV2 |
|---|---|---|---|
| Collect Coal | 0.0% | 0.4% | **14.7%** |
| Collect Diamond | 0.0% | 0.0% | 0.0% |
| Collect Drink | 24.0% | 30.3% | **80.0%** |
| Collect Iron | 0.0% | 0.0% | **0.0%** |
| Collect Sapling | **97.4%** | 66.7% | 86.6% |
| Collect Stone | 0.2% | 3.0% | **42.7%** |
| Collect Wood | 74.9% | 83.0% | **92.7%** |
| Defeat Skeleton | 0.7% | 0.2% | **2.6%** |
| Defeat Zombie | 39.6% | 2.0% | **53.1%** |
| Eat Cow | **26.1%** | 12.0% | 17.1% |
| Eat Plant | 0.0% | 0.0% | **0.1%** |
| Make Iron Pickaxe | 0.0% | 0.0% | 0.0% |
| Make Iron Sword | 0.0% | 0.0% | 0.0% |
| Make Stone Pickaxe | 0.0% | 0.0% | **0.2%** |
| Make Stone Sword | 0.0% | 0.0% | **0.3%** |
| Make Wood Pickaxe | 4.8% | 21.1% | **59.6%** |
| Make Wood Sword | 9.8% | 20.1% | **40.2%** |
| Place Furnace | 0.0% | 0.1% | **1.8%** |
| Place Plant | **94.2%** | 65.0% | 84.4% |
| Place Stone | 0.0% | 1.7% | **29.0%** |
| Place Table | 52.3% | 66.1% | **85.7%** |
| Wake Up | **93.3%** | **92.5%** | **92.8%** |
| Score | 4.3% | 4.6% | **10.0%** |

**Table A.1:** Success rates on Crafter with rewards. Success rates are computed as the fraction of episodes during which the achievement has been unlocked at least once. It is computed across all training episodes within the budget of 1M environment steps. The score is the geometric mean of success rates over all achievements, as described in Section 3.3. Note that the score is computed for each seed separately before averaging over seeds and not the other way around. Numbers within 95% of the best number in each row are highlighted in bold.

# B    SUCCESS RATES WITHOUT REWARDS

| Achievement | Random | RND | Plan2Explore |
|---|---|---|---|
| Collect Coal | 0.0% | **0.1%** | 0.1% |
| Collect Diamond | 0.0% | 0.0% | 0.0% |
| Collect Drink | 9.3% | **52.1%** | 48.7% |
| Collect Iron | 0.0% | 0.0% | 0.0% |
| Collect Sapling | **50.2%** | 34.1% | 25.5% |
| Collect Stone | 0.0% | **0.6%** | 0.5% |
| Collect Wood | 24.4% | **49.6%** | 46.8% |
| Defeat Skeleton | 0.0% | **0.3%** | 0.2% |
| Defeat Zombie | 0.1% | **0.3%** | 0.2% |
| Eat Cow | 0.4% | **0.9%** | 0.7% |
| Eat Plant | 0.0% | **0.0%** | 0.0% |
| Make Iron Pickaxe | 0.0% | 0.0% | 0.0% |
| Make Iron Sword | 0.0% | 0.0% | 0.0% |
| Make Stone Pickaxe | 0.0% | 0.0% | **0.0%** |
| Make Stone Sword | 0.0% | 0.0% | **0.0%** |
| Make Wood Pickaxe | 0.3% | 2.5% | **3.3%** |
| Make Wood Sword | 0.3% | 2.6% | **3.3%** |
| Place Furnace | 0.0% | **0.1%** | 0.0% |
| Place Plant | **44.6%** | 21.4% | 14.0% |
| Place Stone | 0.0% | **0.4%** | 0.3% |
| Place Table | 4.4% | **16.7%** | 16.3% |
| Wake Up | **93.6%** | 7.8% | 47.8% |
| Score | 1.6% | 2.0% | **2.1%** |

**Table B.1:** Success rates on Crafter without rewards. Success rates are computed as the fraction of episodes during which the achievement has been unlocked at least once. It is computed across all training episodes within the budget of 1M environment steps. The score is the geometric mean of success rates over all achievements, as described in Section 3.3. Note that the score is computed for each seed separately before averaging over seeds and not the other way around. Numbers within 95% of the best number in each row are highlighted in bold.

## C  SUCCESS RATES OF HUMAN EXPERTS

| Achievement | Human Experts |
|---|---|
| Collect Coal | 86.0% |
| Collect Diamond | 12.0% |
| Collect Drink | 92.0% |
| Collect Iron | 53.0% |
| Collect Sapling | 67.0% |
| Collect Stone | 100.0% |
| Collect Wood | 100.0% |
| Defeat Skeleton | 31.0% |
| Defeat Zombie | 84.0% |
| Eat Cow | 89.0% |
| Eat Plant | 8.0% |
| Make Iron Pickaxe | 26.0% |
| Make Iron Sword | 22.0% |
| Make Stone Pickaxe | 78.0% |
| Make Stone Sword | 78.0% |
| Make Wood Pickaxe | 100.0% |
| Make Wood Sword | 45.0% |
| Place Furnace | 32.0% |
| Place Plant | 24.0% |
| Place Stone | 90.0% |
| Place Table | 100.0% |
| Wake Up | 73.0% |
| Score | 50.5% |

**Table C.1:** Success rates of human experts on Crafter. The success rates of human experts are computed as the fraction of all 100 recorded games during which the achievement has been unlocked at least once. To compute the score analogously to the artificial agents, we randomly split the 100 games into 5 groups that are treated as the different seeds. We then follow the same procedure as for the artificial agents.

## D  EPISODE REWARD

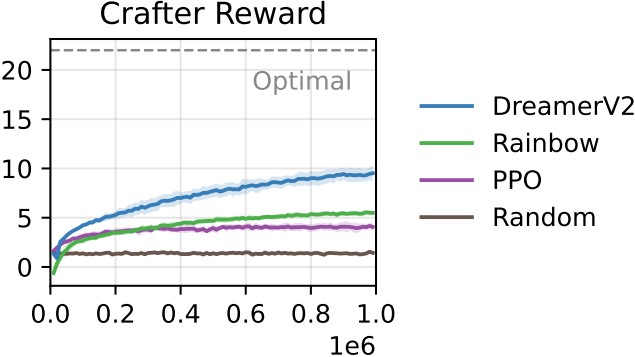

**Figure D.1:** Total episode reward with shaded standard deviation. The optimal achievable episode reward is 22. While visualizing rewards can be informative for debugging, final performance on Crafter should be reported by computing the *score* instead. The score takes the different difficulties of the achievements into account and is defined as the geometric mean of the success rates for all achievements, as described in Section 3.3.

# E    TEXTURES

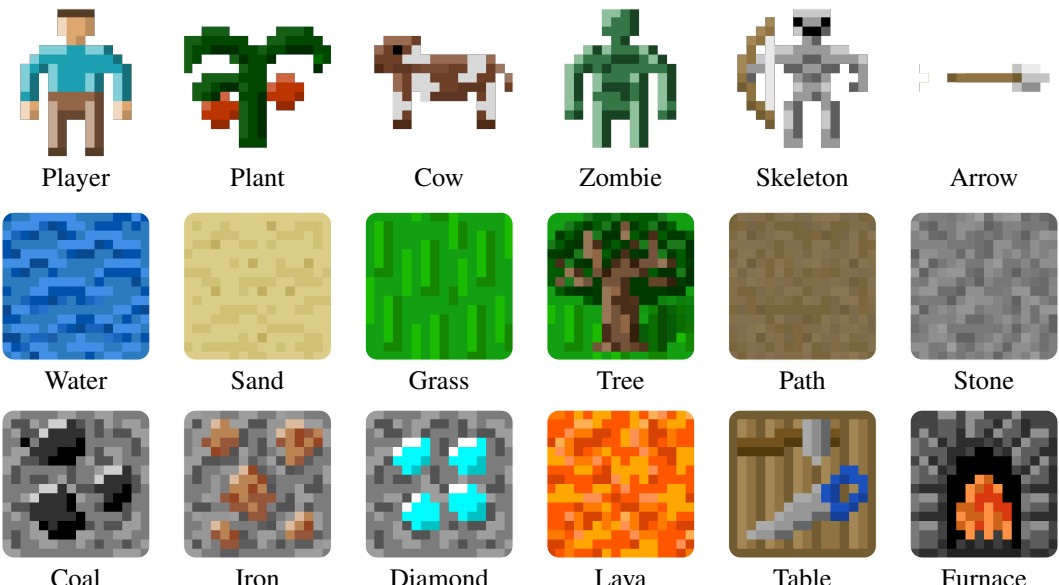

| | | | | | |
|---|---|---|---|---|---|
| Player | Plant | Cow | Zombie | Skeleton | Arrow |
| Water | Sand | Grass | Tree | Path | Stone |
| Coal | Iron | Diamond | Lava | Table | Furnace |

**Figure E.1:** Crafter features worlds with several materials, resources, objects, and creatures. The player can interact with these to collect resources, maintain its food and water supplies, craft pickaxes and swords, and defend itself. The textures were specifically created for Crafter.

# F    ACTION SPACE

| Integer | Name | Requirement |
|---|---|---|
| 0 | Noop | Always applicable. |
| 1 | Move Left | Flat ground left to the agent. |
| 2 | Move Right | Flat ground right to the agent. |
| 3 | Move Up | Flat ground above the agent. |
| 4 | Move Down | Flat ground below the agent. |
| 5 | Do | Facing creature or material; have necessary tool. |
| 6 | Sleep | Energy level is below maximum. |
| 7 | Place Stone | Stone in inventory. |
| 8 | Place Table | Wood in inventory. |
| 9 | Place Furnace | Stone in inventory. |
| 10 | Place Plant | Sapling in inventory. |
| 11 | Make Wood Pickaxe | Nearby table; wood in inventory. |
| 12 | Make Stone Pickaxe | Nearby table; wood, stone in inventory. |
| 13 | Make Iron Pickaxe | Nearby table, furnace; wood, coal, iron an inventory. |
| 14 | Make Wood Sword | Nearby table; wood in inventory. |
| 15 | Make Stone Sword | Nearby table; wood, stone in inventory. |
| 16 | Make Iron Sword | Nearby table, furnace; wood, coal, iron in inventory. |

**Table F.1:** The action space is a flat categorical space, making Crafter easy to use. The 17 actions enable the agent to move, collect materials, place objects, craft objects, and interact with what is in front of the player. Actions whose requirements are not satisfied have no effect.

# G    ACHIEVEMENT CURVES OF RAINBOW

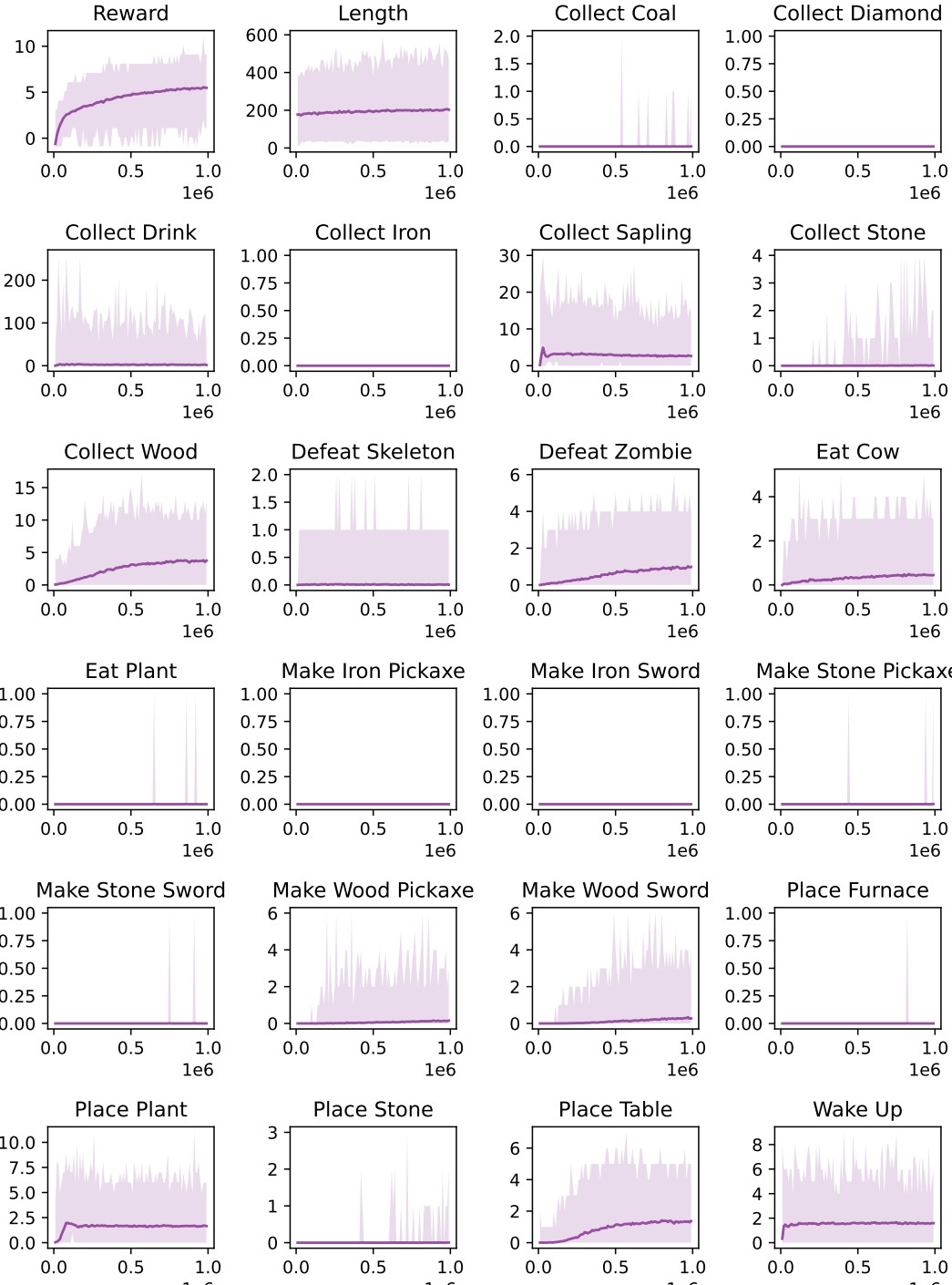

**Figure G.1:** Achievement counts of Rainbow with shaded min and max.

# H ACHIEVEMENT CURVES OF PPO

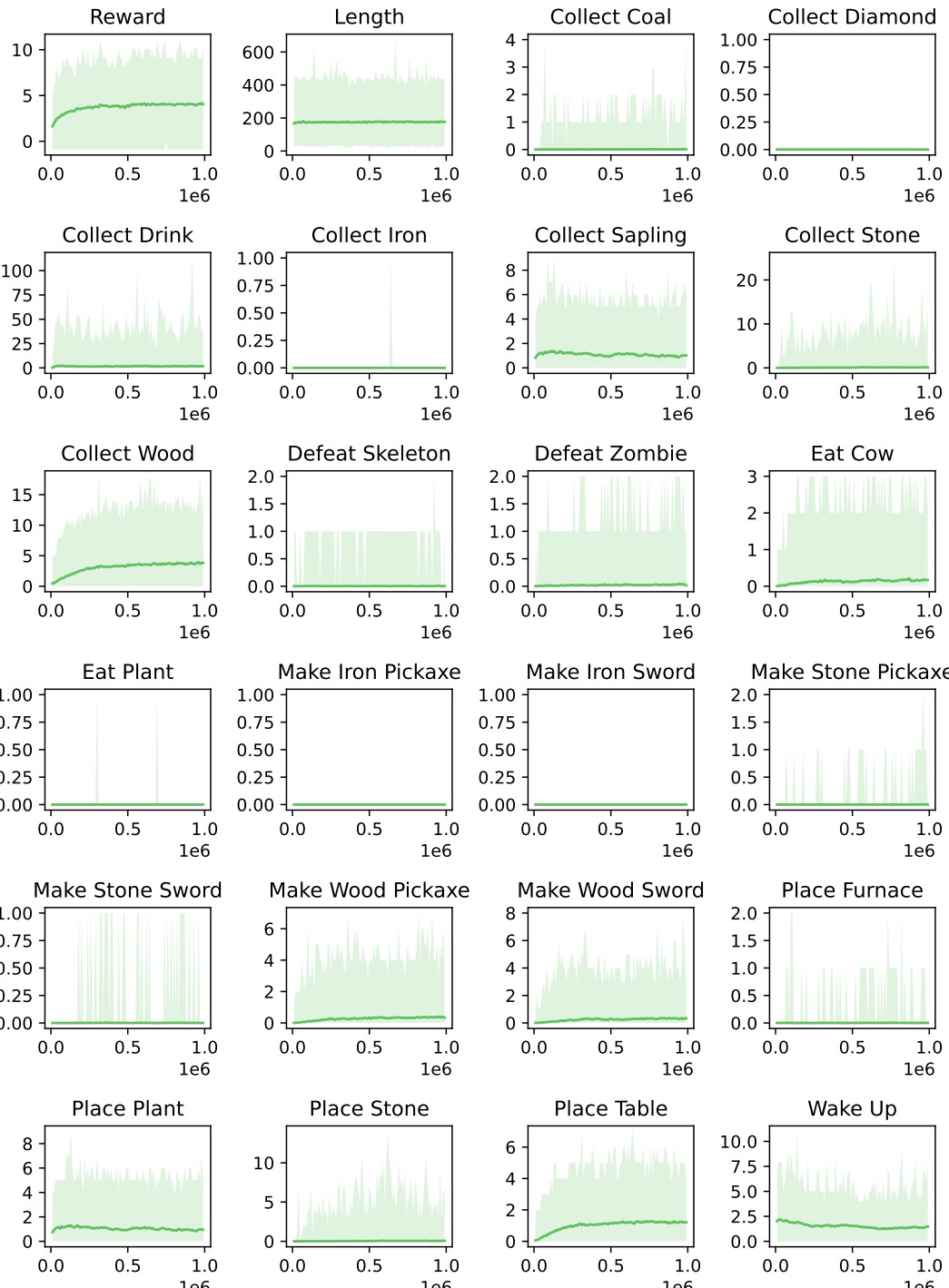

**Figure H.1:** Achievement counts of PPO with shaded min and max.

# I ACHIEVEMENT CURVES OF DREAMERV2

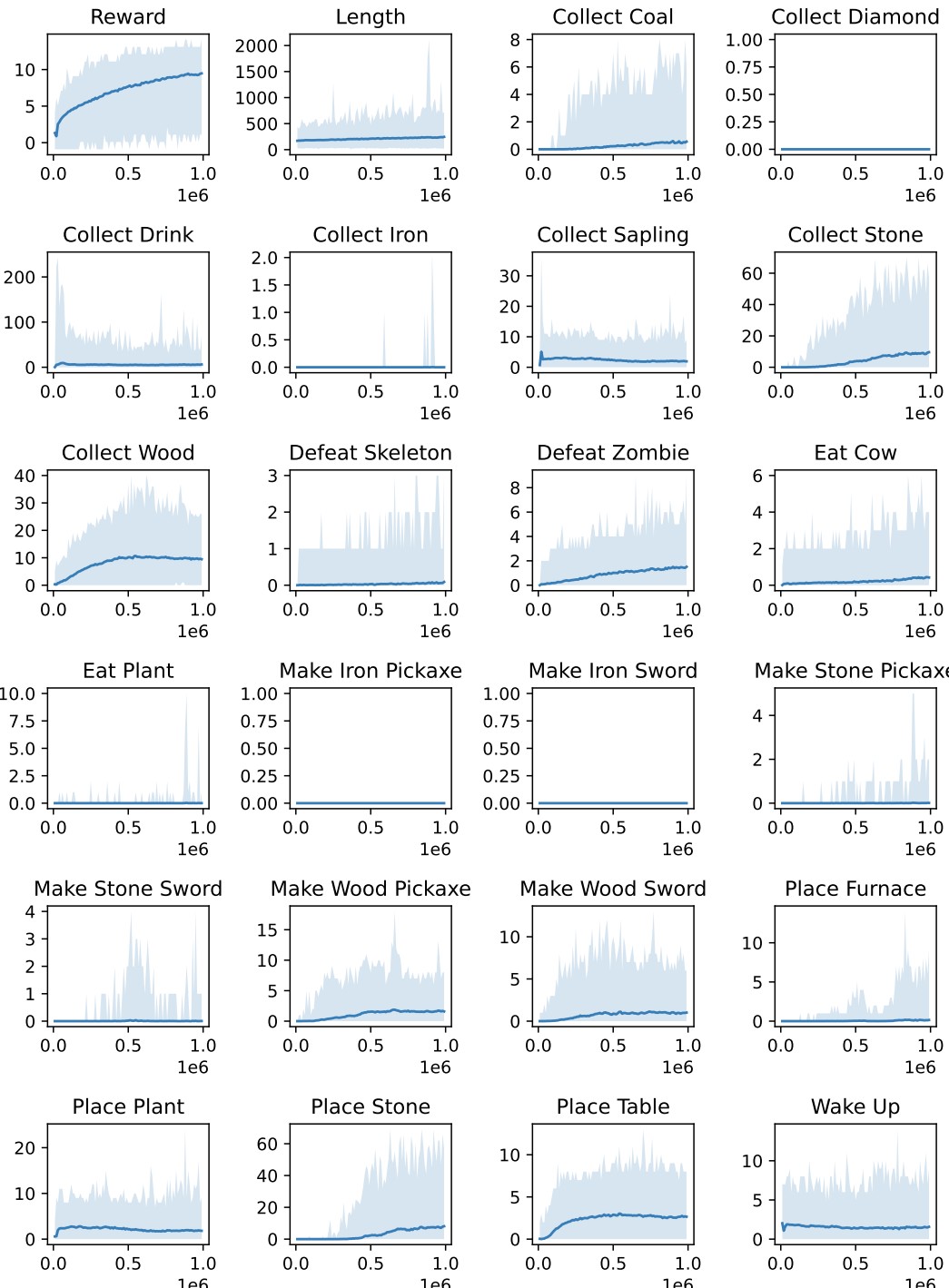

**Figure I.1:** Achievement counts of DreamerV2 with shaded min and max.

## J ACHIEVEMENT CURVES OF RANDOM AGENT

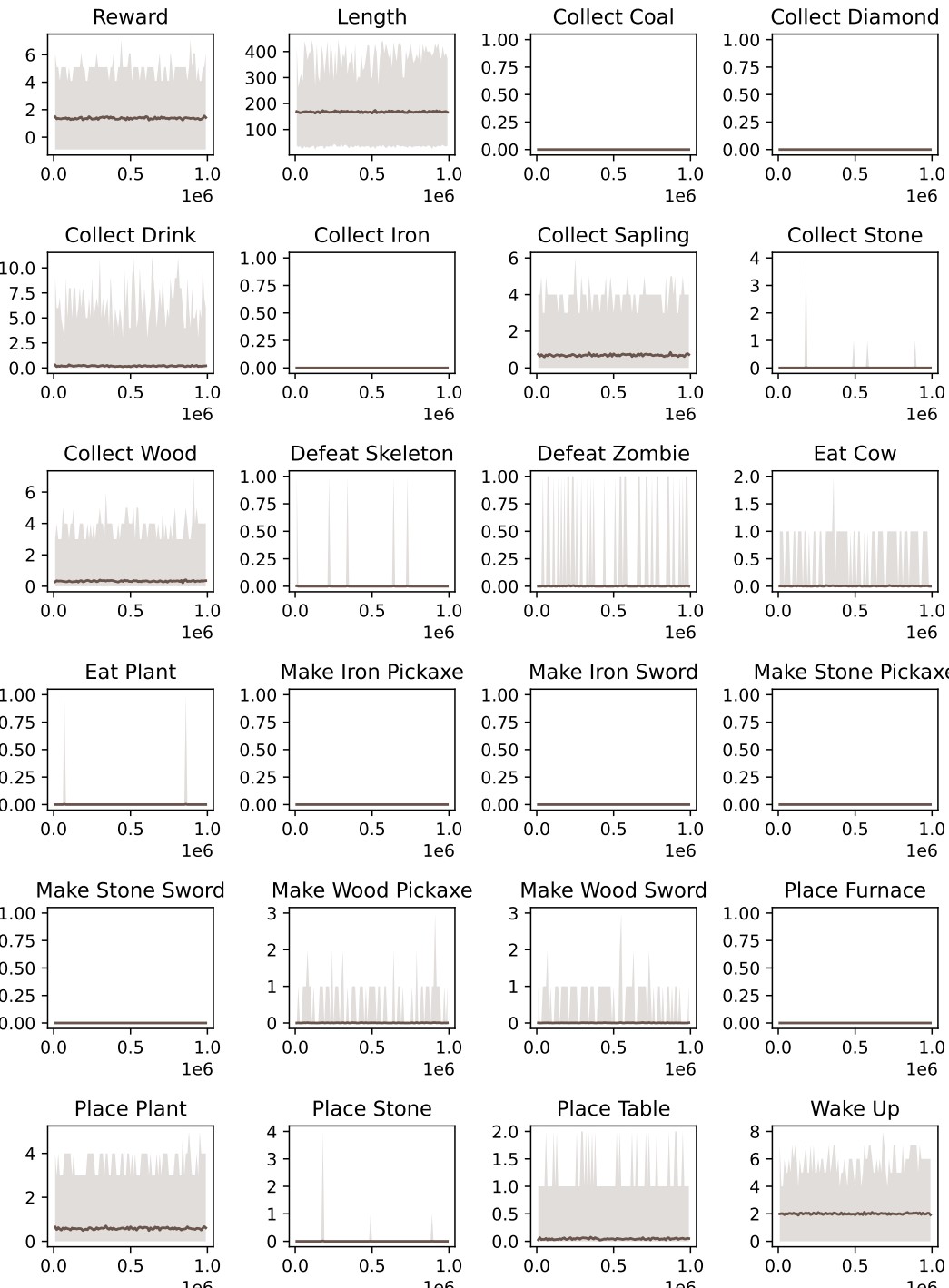

**Figure J.1:** Achievement counts of random actions with shaded min and max.

# K    ACHIEVEMENT CURVES OF UNSUPERVISED RND

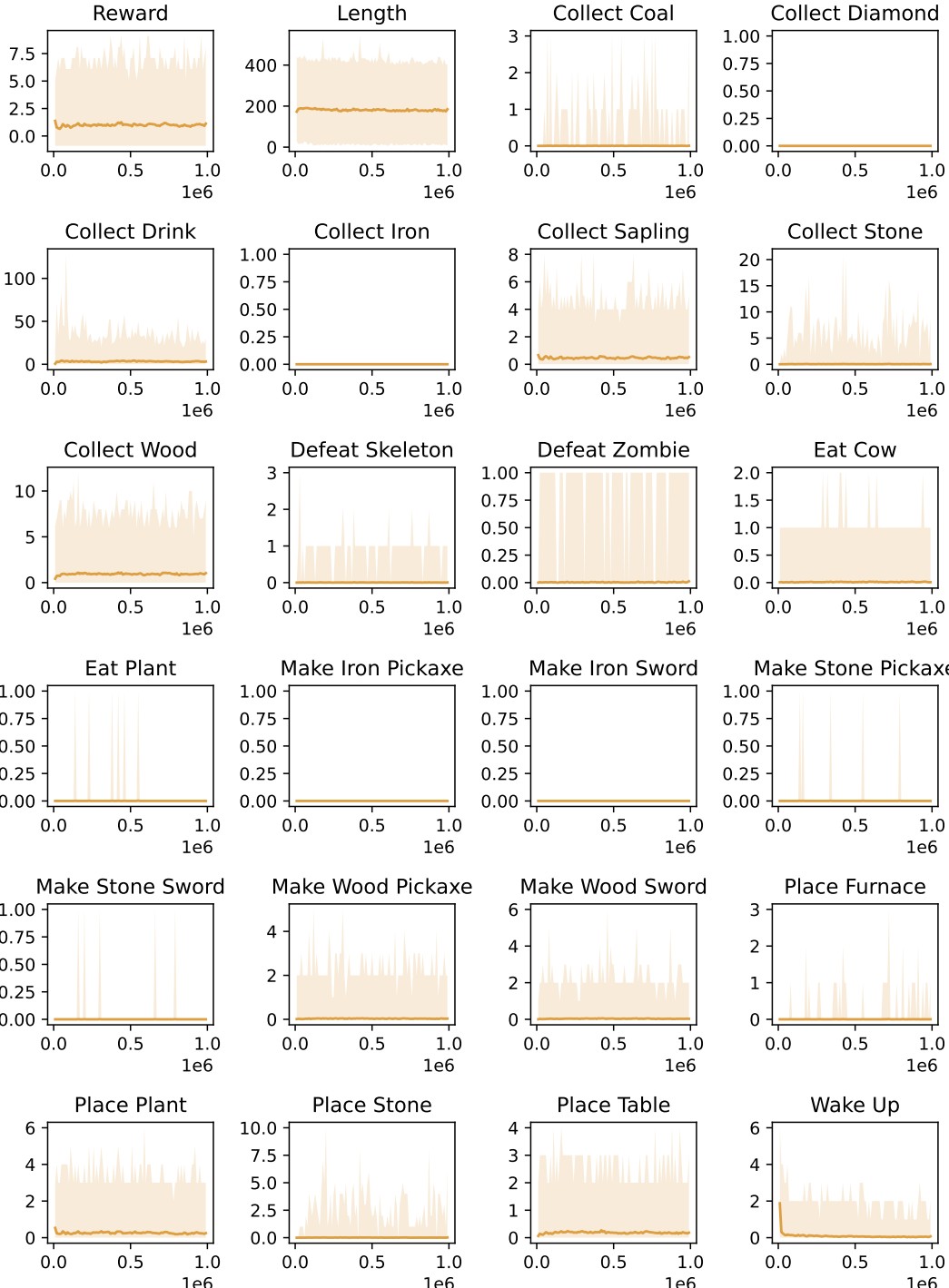

**Figure K.1:** Achievement counts of unsupervised RND with shaded min and max.

# L    ACHIEVEMENT CURVES OF UNSUPERVISED PLAN2EXPLORE

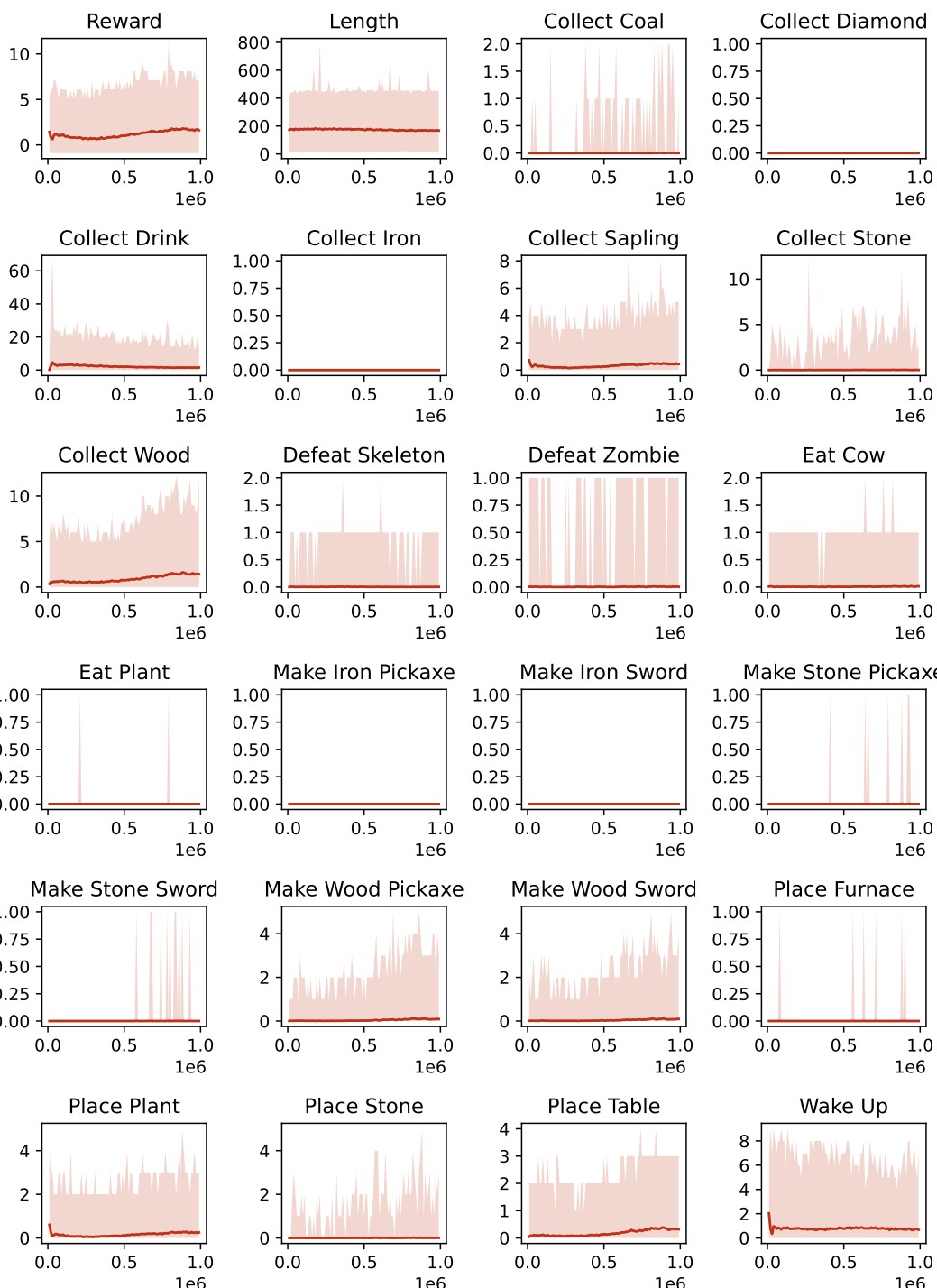

**Figure L.1:** Achievement counts of unsupervised Plan2Explore with shaded min and max.

