# OpenReview forum: "Benchmarking the Spectrum of Agent Capabilities"
_ICLR.cc/2022/Conference — ICLR 2022 Poster_

### Official Review · Reviewer_7FaV · 2021-10-21

**Correctness:** 3
**Technical Novelty And Significance:** 2
**Empirical Novelty And Significance:** 2
**Recommendation:** 6
**Confidence:** 4

**Main Review:**

Strengths:
  - The paper is clearly written, and the evaluation appears sound.
  - The contribution seems useful. The computational demands of MineRL can be challenging especially for researchers with limited compute resource. Having an environment designed specifically for RL in mind is also useful in terms of permitting modifications in the future, and enabling debugging by directly accessing internal state.

Weaknesses:
  - The use case for the benchmark is a bit narrow. It's significantly less complex in terms of both visuals and game mechanics than MineRL, so I expect many researchers to continue to prefer MineRL. Minecraft also has benefits for studying human-robot interaction, as there are already many expert Minecraft players, and it is possible to put an AI directly on a Minecraft server -- something not currently supported by this benchmark.
  - I am concerned that RL algorithms are being trained to maximize the reward but being evaluated on a correlated but distinct metric (aggregate score). It seems conceivable than RL algorithm A could achieve higher reward than algorithm B but a lower score -- how should we interpret such results? In some sense, B produces a stronger policy, but I hesitate to say it is a better RL algorithm (unless you have reason to believe the reward is going to be misspecified in a very particular way).

    While you do report reward in appendix D, I'd suggest either (a) justifying why it's valid to judge RL algorithms based on something other than reward attained; (b) switch to reward as the primary metric.
  - Section 4.3 could be strengthened by including a quantitative description of the agent behaviour (e.g. score attained) in addition to the (very interesting) qualitative description.  In particular, knowing the score would be helpful to contextualize how close DreamerV2 @ 50M timesteps is to expert performance. In general, it's useful to have a sense of if this benchmark is solveable given a 50x increase in timesteps (making it largely about improving sample efficiency of algorithms) or if there still remain some key challenges there (likely requiring a more radical overhaul of RL algorithms to tackle).

Unfortunately the acknowledgements were left in the manuscript which partially de-anonymizes the submission.

Minor points:
  - Section 3.1: "the resources it can collect" -- "it" (for the agent) sounds jarring when you previously refer to agent as "the player" (expect he/she). Consider changing "player" to "agent", or replacing "it" with "the player" or "the agent". This applies to the many subsequent uses of "it" to refer to the player as well.
  - Section 3.1: "diamonds, the player" -> "diamonds, that the player". Same comment as before about "it" and "player".
  - Section 3.1: "Skeletons live in caves and try to keep the player at a distance to shoot arrows at the player." I'm confused by this, are you saying the skeletons shoot players if they move within a certain range? (Perhaps should be "by shooting" rather than "to shoot".) Or are the skeletons moving, trying to keep the player at a distance where the player can be shot?
  - Section 3.1: "The player can interact with creatures to decrease their health points." Why would they want to decrease their health points? If they don't, but it's sometimes difficult to avoid, consider using a word other than "can" (which implies possibility not necessity).
  - Section 4.1: "lack behind the score" -> "lag behind ..."? Similar comment in 4.3: "lack far behind optimal performance".
  - Section 4.1: "visualized in Figure 7" -- I think you mean Figure 6? I think 4.2 also should cite Figure 7 instead of Figure 6 (i.e. swap them.)

**Summary Of The Paper:**

The work introduces a gridworld-style benchmark inspired by Minecraft. It is similar in style to MineRL, but is significantly more computationally efficient, and easier to use (can be run directly without requiring e.g. an external Java application). Despite this simplification, it proves to be a challenging benchmark, with state-of-the-art RL algorithms achieving no more than 10% average success rate across tasks (e.g. make a pickaxe, collect food, find a diamond).

**Summary Of The Review:**

The benchmark developed will be valuable to those seeking to test agents in Minecraft-like environments but without sufficient computational resources to test at scale in MineRL. However, this use case is somewhat narrow, and additionally the choice of evaluation metric (aggregate score) is not well justified (especially in situations where it may diverge from the reward). I would assess the current submission as being borderline, but I lean slightly towards reject. I would consider increasing my score if the evaluation criteria were more clearly justified.

Update: I have increased my score to marginal accept after the author's explanation of the choice of GM, and their inclusion of returns in the main paper.

---

> ### Author Response · Authors · 2021-11-12
> **Response to Reviewer 7FaV**
>
> Thank you for your detailed review! Below, we address your concerns by clarifying the synergy with Minecraft, adding an explanation of the motivation and significance of the aggregated score, and adding the quantitative result for Sec 4.3.
>
> Please confirm whether our response fully addresses your concerns or whether there are any remaining concerns that motivate recommending rejection, which we will be more than happy to address.
>
> > Despite this simplification, it proves to be a challenging benchmark, with state-of-the-art RL algorithms achieving no more than 10% average success rate across tasks (e.g. make a pickaxe, collect food, find a diamond).
>
> Exactly! Learning sophisticated long-horizon behaviors is challenging for current methods even in the absence of 3D perception and navigation. Crafter allows researchers to study this challenge separately, with the goal of progressing faster towards solving environment such as Minecraft in the future.
>
> > The use case for the benchmark is a bit narrow. It's significantly less complex in terms of both visuals and game mechanics than MineRL, so I expect many researchers to continue to prefer MineRL.
>
> The goal of Crafter is not to replace Minecraft but to progress faster towards it! As you pointed out, Crafter *"proves to be a challenging benchmark, with state-of-the-art RL algorithms achieving no more than 10% average success rate".* This shows that learning complex long-term behaviors is an open research problem even in the absence of more items and 3D worlds. Crafter aims to accelerate research on this problem through substantially increased iteration times compared to Minecraft.
>
> > I am concerned that RL algorithms are being trained to maximize the reward but being evaluated on a correlated but distinct metric (aggregate score). It seems conceivable than RL algorithm A could achieve higher reward than algorithm B but a lower score -- how should we interpret such results? In some sense, B produces a stronger policy, but I hesitate to say it is a better RL algorithm (unless you have reason to believe the reward is going to be misspecified in a very particular way).
>
> That's a good point. Reporting aggregate performance across multiple tasks is common across the literature, for example with the human-normalized median score on the Atari suite. Even though Atari agents do not directly optimize the median score, it is still used as a more meaningful metric than the average. This is because the average is dominated by a small number of easy games. The same is true for Crafter. In both benchmarks, the optimal expected reward also yields the optimal aggregate score, but for suboptimal agents, the aggregate score is more robust to outliers.
>
> When developing the environment, we initially only measured the sum of rewards. However, we found that agents made vastly different progress in the game while showing similar rewards, because of the broad spectrum of abilities measured by Crafter. On the other hand, we did not want to guess how difficult each task might be and assign different rewards based on that, which might also differ between algorithms.
>
> The geometric mean is an elegant solution to this dilemma of aggregating scores of multiple tasks. It automatically weighs difficult tasks more strongly, without having to know a priori which tasks are difficult. We see this as a step towards better multi-task evaluation not just in Crafter but also in more complex environments that will be developed in the future. We will add the above discussion to Section 3.3.
>
> > Section 4.3 could be strengthened by including a quantitative description of the agent behaviour (e.g. score attained) in addition to the (very interesting) qualitative description. In particular, knowing the score would be helpful to contextualize how close DreamerV2 @ 50M timesteps is to expert performance.
>
> Thank you for the suggestion! The reward achieved using 50M timesteps is ~15. This means that there are key challenges for current methods, in addition to increasing sample efficiency. We did not log the individual success rates for this experiment, but we will rerun the experiment and add the result to the paper and also add the code to the release for reproducibility.

---

> > ### Comment · Reviewer_7FaV · 2021-11-13
> > **Clarification**
> >
> > Thanks for your response, this has addressed most of my questions.
> >
> > I don't quite understand your justification for the score:
> >
> > > That's a good point. Reporting aggregate performance across multiple tasks is common across the literature, for example with the human-normalized median score on the Atari suite. Even though Atari agents do not directly optimize the median score, it is still used as a more meaningful metric than the average. This is because the average is dominated by a small number of easy games. The same is true for Crafter. In both benchmarks, the optimal expected reward also yields the optimal aggregate score, but for suboptimal agents, the aggregate score is more robust to outliers.
> >
> > In the Arcade Learning Environment, separate agents are normally trained for each of the tasks. The median reward acts as a robust measure of the RL algorithm performance. But maximizing the reward for each subtask is the best way of getting a high median score: it is always better to have a higher reward at a given task, and since the policies and training for each task are independent, there is no trade-off between rewards attained at different tasks.
> >
> > By contrast, in this setting you're training a single agent, which acquires reward by solving a series of tasks. While I agree the *optimal* solution also achieves a perfect score, a suboptimal agent A could achieve lower reward than agent B but a higher aggregate score. It may be that in practice this is rarely a problem in this setting -- but that isn't clear to me from the results in the paper.

---

> > > ### Author Response · Authors · 2021-11-16
> > > **Response to Reviewer 7FaV**
> > >
> > > Thank you for your fast response and for confirming that we addressed most of your questions! Below, we address your remaining concern by including the expected return as part of the benchmark and by elaborating on the motivation for the geometric mean score, its relation to the expected return, and its value for unsupervised RL.
> > >
> > > > in this setting you're training a single agent, which acquires reward by solving a series of tasks. [...] a suboptimal agent A could achieve lower reward than agent B but a higher aggregate score.
> > >
> > > After further consideration, we agree with you that the expected return is a valuable metric because reward-based agents directly optimize for it. Therefore, we will report both the geometric mean and the expected return. In addition to the expected return curves that are already in the appendix, we will add a table of expected returns to the main text and highlight it as part of the benchmark. We additionally elaborate on the significance and interpretation of the geometric mean, as summarized below.
> > >
> > > **Difference between expected return and geometric mean**
> > >
> > > Let's consider a partially-trained agent with a certain expected return and geometric mean score. If the agent improves on any one of the tasks, without forgetting the tasks it already solves, then it will improve on both expected return and geometric mean. However, if an agent does not have enough capacity or non-stationary learning, it has to decide how much focus to give each task.
> > >
> > > In this situation, the expected return favors improving easy tasks over improving difficult tasks, because this increases reward while requiring less capacity. As a result, the expected return is dominated by a small number of easy tasks, as is evident for example in Atari. The geometric mean puts more emphasis on improving difficult tasks, because they are weighted stronger in the aggregation.
> > >
> > > For example, improving from never collecting a diamond to collecting it 1% of the time is a substantial improvement, whereas improving from collecting wood 95% to 96% of the time is less of an accomplishment. In the expected mean, the small improvement from 0% to 1% would likely be hidden by all the small performance changes on easy tasks.
> > >
> > > **Geometric mean for unsupervised RL**
> > >
> > > In unsupervised RL, we often prefer agents to learn a broad range of behaviors over repeating a small number of easy behaviors over and over. Unsupervised agents can also get bored of behaviors after a while and move on to new behaviors.
> > >
> > > For example, a good agent demonstrates many behaviors at some time throughout learning, resulting in collecting wood 20% of the time, drinking 20% of the time, and collecting diamonds 5% of the time. A poor agent that focuses exclusively on easy tasks collects wood 40% of the time, drinks 40% of the time, but never finds a diamond. Across those three tasks, the good agent achieves an expected return of 0.45 and a geometric mean success of 12.8% whereas the poor agent achieves an expected return of 0.8 and a geometric mean success of 10.9%.
> > >
> > > The more difficult tasks are available in the environment, the more important the geometric means becomes for evaluating unsupervised agents compared to the expected return. Thus, the geometric mean will likely gain in significance as the capabilities of unsupervised RL methods grow and we move towards more and more complex environments.
> > >
> > > Please let us know whether we have addressed all your concerns that were holding you back from recommending acceptance, or whether there are any remaining issues, which we would be happy to address!

---

> > > > ### Comment · Reviewer_7FaV · 2021-11-19
> > > > **Thanks for the explanation**
> > > >
> > > > Thanks for the response. I agree that the geometric mean is a reasonable measure of performance of unsupervised RL in this benchmark, and likely a more reasonable one than the return. It feels a bit strange as with unsupervised RL we care less about the performance of the policy per-se, and more what one can do with it e.g. by fine-tuning the learned representation on a particular task. But I agree the diversity of tasks it (sometimes) solves is correlated with that.
> > > >
> > > > If the paper is revised to include a table of expected returns and a (can be brief, e.g. one paragraph) discussion of the GM score (justification similar to that given above, plus caveats) then I would be happy to increase my vote.

---

> > > > > ### Author Response · Authors · 2021-11-19
> > > > > **Paper updated**
> > > > >
> > > > > Thank you! We have updated the paper to include the episode returns in Table 1 and added a discussion paragraph to the end of Section 3.3 Evaluation Protocol. The revision is available on OpenReview now. If any issues remain, please let us know.

---

> > > > > ### Author Response · Authors · 2021-11-23
> > > > > **Reminder for Reviewer 7FaV**
> > > > >
> > > > > Dear reviewer, we've updated the paper with your suggested changes a couple of days ago. Would you mind increasing your score as mentioned, or suggest any further changes to the paper if our updated didn't capture what you had in mind? Thanks!

---

> > > > > > ### Comment · Reviewer_7FaV · 2021-11-23
> > > > > > **Updated**
> > > > > >
> > > > > > Thanks for the nudge, I have updated the score and original review now.

---

> > > > > > > ### Author Response · Authors · 2021-11-23
> > > > > > > **Thanks**
> > > > > > >
> > > > > > > Thank you!

---

### Official Review · Reviewer_ix7A · 2021-10-25

**Correctness:** 4
**Technical Novelty And Significance:** 3
**Empirical Novelty And Significance:** 2
**Recommendation:** 5
**Confidence:** 4

**Main Review:**

Strengths: This environment has a credible case that it could help accelerate research on exploration and generalization. From the perspective of generalization, its great that each new episode comes with an entirely new, procedurally generated map. Additionally, skeletons and antagonistic enemies introduce additional complexity to the survival task, as does the limited supply of food/water/energy. The existing baselines indicate that the task is neither too hard nor too easy for existing RL agents.

Weaknesses: Let us imagine the world in which a significant portion of the RL research community started focusing on the Crafter benchmark. Would this shift in focus lead to the development of better RL agents along the dimensions of exploration/generalization/reusable-skills/credit-assignment/memory? Would these new agents be generalizable beyond Crafter to other domains and applications?

I see a possible failure case of agents that are generalizable only in the senses that Crafter requires - e.g. able to handle new maps, but perhaps are non-generalizable beyond Crafter. E.g. agents which may understand the specific crafting tree and strategies for acquiring food and shelter that are specific to Crafter. Other environments like ProcGen combat this problem by having entirely separate held-out environments which discourage the agent from overfitting to specifics of the training environments. Crafter seems to lack the ability to change its crafting tree and environment dynamics.

Another weakness is the fact that this domain is crafted for the purpose of training RL agents - which means that unlike other domains that focus on solving games designed for humans like Atari and NetHack, Crafter may not be inherently interesting or challenging for humans. As a community, would we be impressed by agents that could complete the full set of achievements on this domain? It's unclear, but it's doubtful that anyone would be able to relate to the difficulty of Crafter without having first used it as a research framework. Contrast this to a domain like NetHack / StarCraft2 / DoTA / Chess+Go - in which players are already familiar and can relate to the difficult of solving.

I would also be interested to understand how easy or difficult it is to hand-code a heuristic agent to solve this task. I think it would be somewhat less interesting as a task if it was possible to create a heuristic agent that could solve it using only a few hundred lines of code.

In summary, it's clear that significant thought and work has gone into creating the Crafter environment. There interesting aspects of the environment that are novel such as the need to find food/water/shelter/sleep. Episodes are long and maps are procedurally generated so there is certainly a possibility that Crafter may be applicable to continuous/lifelong learning agents. However, there are still limitations in terms of the environment being relatively fixed/static at the meta level of the crafting tree and ways of interacting with different types of objects/enemies. This makes it unclear whether agents and algorithms developed for or trained in Crafter will be general beyond this particular environment.

Update after author response: I'm still borderline on this paper. I'd love to hear from someone who works on RL in Minecraft/MineRL as to whether this environment would be welcome and useful in the space.

**Summary Of The Paper:**

This paper introduces a new environment for development of agent capabilities, called Crafter. The environment is procedurally generated and consists of a 2-D world inhabited by various resources, terrain types, and objects. The agent is rewarded for crafting items and accomplishing achievements from a set of 22 possible achievements. The objective of this environment is to encourage RL research on the topics of exploration, generalization, reusable skills, credit assignment, and memory. Benchmark results are presented for various RL algorithms under a reward and no-reward settings, as well as human experts. In general, the current RL methods fall far short of human performance, although Dreamer-V2 shows emergent behaviors.

**Summary Of The Review:**

It's hard to objectively referee this type of a paper because as with many new environments, beauty is in the eye of the beholder. Overall, I'm not particularly excited by another 2D gridworld-style environment - but I will admit that Crafter has several interesting aspects that aren't featured in other environments. Similarly, I'm moderately interested by the emergent behaviors demonstrated by the Dreamer-V2 agent, but not blown away (unlike emergent behaviors in OpenAI Hide and Seek for example). So overall I'm lukewarm on Crafter - I could see it possibly appealing to a subset of the RL community and perhaps helping to drive research in particular subfields of RL, but this is just my guess. Should it be accepted as a conference paper? Perhaps, but it may be better promoted through other means like NeurIPS competitions, blog posts, and similar.

---

> ### Author Response · Authors · 2021-11-12
> **Response to Reviewer ix7A (Part 2)**
>
> > I would also be interested to understand how easy or difficult it is to hand-code a heuristic agent to solve this task. I think it would be somewhat less interesting as a task if it was possible to create a heuristic agent that could solve it using only a few hundred lines of code.
>
> That's an interesting idea. This is certainly not trivial but may be possible with a large enough time investment, similar to how NetHack could likely be solved with a very sophisticated planner that uses complete domain knowledge of the environment. Note that this again applies to many RL envs, such as Atari and MuJoCo. An additional challenge in Crafter would be to extract semantic representations from the images, with the visuals continuously changing with the day/night cycle.
>
> > However, there are still limitations in terms of the environment being relatively fixed/static at the meta level of the crafting tree and ways of interacting with different types of objects/enemies. This makes it unclear whether agents and algorithms developed for or trained in Crafter will be general beyond this particular environment.
>
> We tuned the difficulty of Crafter to be appropriate for current RL research. As you said, *"The existing baselines indicate that the task is neither too hard nor too easy for existing RL agents."* Moreover, ensured that human players can solve the game during at least a fraction of games. That said, Crafter is written purely in Python and can easily be extended by additional items, resources, enemies, and achievements as the research community progresses towards solving the challenges that are already present.
>
> > I'm moderately interested by the emergent behaviors demonstrated by the Dreamer-V2 agent, but not blown away (unlike emergent behaviors in OpenAI Hide and Seek for example).
>
> It is worth pointing out that while OpenAI Hide and Seek was an interesting project, it was also marketed well. Despite the 3D visualizations, their RL agents learned only from low-dimensional range finders and ground truth XY positions of all agents and objects. The environment also only included agents and shapes. In comparison, the emergent behaviors we find in Crafter are learned completely end-to-end from pixels in a diverse environment that gives a lot of freedom to the agent. For example, the agent learns to dig caves and close them for the night despite all the other possible resources and enemies it could interact with instead.

---

> ### Author Response · Authors · 2021-11-12
> **Response to Reviewer ix7A (Part 1)**
>
> Thank you for your detailed review! Below, we address your concerns by discussing domain-specific knowledge, comparing the evaluation setup to ProcGen, discussing the pros/cons compared to established games that were not designed for research, and highlighting the extensibility of the environment.
>
> Please confirm whether our response fully addresses your concerns or whether there are any remaining concerns that motivate recommending rejection, which we will be more than happy to address.
>
> > I see a possible failure case of agents that are generalizable only in the senses that Crafter requires [...] E.g. agents which may understand the specific crafting tree and strategies for acquiring food and shelter that are specific to Crafter.
>
> To understand an aspect of the environment, agents either need to learn it from experience (as we want them to) or researchers could specify this prior knowledge when designing the algorithm. Any environment could be gamed by making strong domain-specific assumptions. However, many RL researchers aim to design algorithms that do not make strong domain-specific assumptions that are more widely applicable.
>
> > Other environments like ProcGen combat this problem by having entirely separate held-out environments which discourage the agent from overfitting to specifics of the training environments. Crafter seems to lack the ability to change its crafting tree and environment dynamics.
>
> Both ProcGen and Crafter report scores that are computed on new randomly generated maps. A small difference is that ProcGen uses a fixed set of evaluation seeds, while Crafter evalues on randomly selected seeds. Training on the experience after the fact is not a problem, because an agent will not be evaluated on the same situation again in the future.
>
> > Another weakness is the fact that this domain is crafted for the purpose of training RL agents - which means that unlike other domains that focus on solving games designed for humans like Atari and NetHack, Crafter may not be inherently interesting or challenging for humans.
>
> We recorded a dataset of 5 human players who practiced for several hours beforehand. Their Crafter score of 50.5% shows that the environment is indeed challenging for humans. Moreover, there was substantial learning progress visible among human players over the course of several hours of play. Please feel free to try out the commands in Fig 2 to play the game yourself. Interestingness is of course subjective, but we enjoyed playing the game and were challenged by it.
>
> > As a community, would we be impressed by agents that could complete the full set of achievements on this domain? It's unclear, but it's doubtful that anyone would be able to relate to the difficulty of Crafter without having first used it as a research framework. Contrast this to a domain like NetHack / StarCraft2 / DoTA / Chess+Go - in which players are already familiar and can relate to the difficult of solving.
>
> Using established games as RL benchmarks has the benefit of strong human baselines, and those games have a place in the research community. However, many researchers arguably still have no good intuitions about their challenges without playing the games. Crafter offers clear benefits over many of those games:
>
> - Existing games often pose a small number of challenges. For example, Chess and Go are fully observed MDPs and cannot evaluate memory abilities. NetHack uses purely symbolic inputs and cannot evaluate representation learning. Crafter is specifically designed for comparing RL agents across a broad range of abilities.
>
> - Many of the mentioned games, such as StarCraft and Dota, are extremely resource intensive. The required training infrastructure and compute cost makes them  infeasible as benchmarks for the academic community. Crafter allows for meaningful evaluation within 1M environment frames, resulting in fast iteration times.
>
> - Those established games were not designed for research and are often difficult to set up, use, and modify. For example, Atari ROMs are not available under a free license. Minecraft requires a JVM and running window server. The StarCraft (pysc2) instructions require to separately set up the actual game, the maps, and the Python bindings.

---

> > ### Comment · Reviewer_ix7A · 2021-11-12
> > **On the fixed crafting tree**
> >
> > > To understand an aspect of the environment, agents either need to learn it from experience (as we want them to) or researchers could specify this prior knowledge when designing the algorithm. Any environment could be gamed by making strong domain-specific assumptions. However, many RL researchers aim to design algorithms that do not make strong domain-specific assumptions that are more widely applicable.
> >
> > I guess the problem is enforcing/encouraging that researchers not encode extensive Crafter-specific prior knowledge in their algorithms. Speaking from experience reviewing and authoring RL environments - I can say with confidence that researchers will submit and possibly publish any algorithms that improve SOTA performance on Crafter, regardless of the amount of prior knowledge their agents include. The fight for new high scores introduces a slippery slope that encourages researchers to take as much advantage as possible of the environment. It introduces the subsequent problem of how to compare agents that get high scores on Crafter but use lots of domain specific knowledge versus those that get lower scores without domain specific knowledge.
> >
> > > Any environment can be gamed by making strong domain-specific assumptions.
> >
> > I think that some environments are more susceptible to being gamed than others, and there are approaches that can be taken to make environments harder to game. For example, Atari is hard to game because of the constraint that a single agent + hyperparameter setting needs to do well across many different games. The ProcGen competition at NeurIPS was hard to game because they evaluated agents on heldout games - entire environments never seen during training. Crafter's random map generation makes it harder to game than if it had a fixed map. But the fixed crafting tree, fixed challenges, and fixed dynamics make it susceptible to being gamed at this meta-level.

---

> > > ### Author Response · Authors · 2021-11-16
> > > **Response to Reviewer ix7A**
> > >
> > > Thank you for your quick response!
> > >
> > > Below, we address your remaining concerns by clarifying that Crafter is not meant to be used as the sole benchmark for future algorithm papers, discussing how Crafter shares the property of fixed hyper parameters across all tasks with Atari that makes it harder to game, arguing that secret holdout environments are suitable for sponsored competitions but not for open source environments, and elaborating on why we believe a normal fixed technology tree is sufficient and preferable.
> > >
> > > > The fight for new high scores introduces a slippery slope that encourages researchers to take as much advantage as possible of the environment.
> > >
> > > While people might develop algorithms that are specific to Crafter and not generally applicable, we do not see this as a major problem. The research community will most likely not build upon such specialized algorithms.
> > >
> > > Moreover, Crafter is not intended to replace all other RL benchmarks. In practice, RL papers are expected to evaluate new algorithms on more than one environment. This is still the case with Crafter, even though its multi-task evaluation allows for more confidence in the performance of an algorithm than a single-task environment would.
> > >
> > > > Atari is hard to game because of the constraint that a single agent + hyperparameter setting needs to do well across many different games.
> > >
> > > Fixing the hyper parameters across many environments makes the Atari suite harder to game than it would be without this constraint. The same mechanism is present in Crafter. Namely, Crafter is harder to game because one agent with one set of hyper parameters is evaluated across all 22 tasks, which measure different skills (e.g. reflexes, foraging, credit assignment, memory). In addition, it is harder to break the convention of using the same hyper parameters in Crafter, where only a single agent is trained, than in Atari, where many separate agents are trained. For example, see Table 1 in the MuZero paper (Schrittwieser et al. 2020), where UNREAL tuned the hyper parameters separately for each game.
> > >
> > > > The ProcGen competition at NeurIPS was hard to game because they evaluated agents on heldout games - entire environments never seen during training.
> > >
> > > Using a holdout set of environments is a great idea for an organized competition. However, keeping the evaluation environments secret requires compute infrastructure to evaluate everyone's submissions. Thus, it is practical for sponsored competitions but less for open source environments. That said, as mentioned earlier, Crafter computes the scores on randomly generated and unseen episodes, making it significantly harder to game than environments without procedural generation.
> > >
> > > > Crafter's random map generation makes it harder to game than if it had a fixed map. But the fixed crafting tree, fixed challenges, and fixed dynamics make it susceptible to being gamed at this meta-level.
> > >
> > > We see it as natural that an RL environment poses fixed challenges and dynamics. Episodes are required to share mutual information with each other for learning across episodes to be possible.
> > >
> > > The technology tree is fixed like in Minecraft. We do not see it as a problem that a subset of researchers might be interested in algorithms that are specific to a fixed technology tree. Such abstract milestones are available for some application domains of RL, for example when we want a robot to follow a recipe to cook a meal. This makes Crafter an interesting environment also for the robotics community. We are optimistic that the RL community will recognize the difference between methods that require such additional knowledge − and thus operate under a more limited problem setting − and those that don't. While one could easily randomize the technology tree between episodes, Crafter is already challenging enough for current RL methods. Moreover, the environment is written purely in Python and can be easily modified by the community in this way to create a new benchmark if desired.
> > >
> > > Please confirm whether we have addressed the issues you raised satisfactorily and you are recommending acceptance, or whether there are any remaining major concerns, which will be more than happy to address!

---

> > > > ### Author Response · Authors · 2021-11-19
> > > > **Reminder for Reviewer ix7A**
> > > >
> > > > Dear reviewer, with the discussion period coming to an end, could you please confirm whether our response above has addressed your main concerns and whether you are planning to update your recommendation, or whether there are any remaining specific concerns and how we could address them? Thanks!

---

> > > > > ### Comment · Reviewer_ix7A · 2021-11-23
> > > > > **Response to Reminder for Reviewer ix7A**
> > > > >
> > > > > Dear authors, my concerns seem to have largely dismissed in the response above. At this point I think it's accurate to say we have a difference of opinion regarding the potential for agents to game the environment, and the implications for downstream research in the environment.
> > > > >
> > > > > At a meta-level I appreciate the proactive approach taken by the authors to try to address reviewer concerns, but asking for repeated confirmation about whether the responses have "addressed all the issues necessary for recommending acceptance" feels borderline aggressive. If you feel that I am out of line here, I encourage you to raise the issue with the meta-reviewer/area-chair.

---

> > > > > > ### Author Response · Authors · 2021-11-23
> > > > > > **Response to Reviewer ix7A**
> > > > > >
> > > > > > Dear reviewer, thank you for your response!
> > > > > >
> > > > > > We apologize for being perceived as a bit aggressive, that was not our intention. We just wanted to keep the discussion focused on the main points and make sure we don't miss any of your remaining concerns, as there is still time to discuss them further.
> > > > > >
> > > > > > We agree that we perhaps have different perspectives on how easily the environment can be gamed without being noticed and whether applications of the environment outside of the proposed benchmark setting are a problem or if such creative uses should even be encouraged.
> > > > > >
> > > > > > That said, we would be happy to hear your concrete ideas for improving the environment if you can think of any. If we agree that they are feasible and would improve the environment, we would be happy to realize them.

---

> > > ### Author Response · Authors · 2021-11-23
> > > **Reminder for Reviewer ix7A**
> > >
> > > Dear reviewer, we wanted to send you a second reminder. Could you please acknowledge our response to your concerns and let us know whether you're planning on increasing your score or whether there are any remaining larger issues that need to be addressed? Thank you very much!

---

### Official Review · Reviewer_cMdw · 2021-10-26

**Correctness:** 4
**Technical Novelty And Significance:** 2
**Empirical Novelty And Significance:** 2
**Recommendation:** 5
**Confidence:** 4

**Details Of Ethics Concerns:**

- The paper includes instructions on how to install the Crafter environment (see Figure 2) using the Python pip system. I went to the pip page for Crafter and it had a link to the project web page which revealed the project page, including author's identity and affiliations, blog post about Crafter, github page, link to arxiv with a pre-print (with author identified) and twitter post announcing the release. Furthermore, the paper also identifies other research colleagues in the acknowledgments section.
- In the author's defence they also include an anonymous web page for downloading source code etc.
- I'm not making a judgement here as I'm not really sure what the double-blind review policy is for ICLR. I guess this is a question for the ICLR program chairs.

**Main Review:**

# Strengths
- New RL environments such as this are always welcome.
- The paper is well written and well organised.
- The motivation for the work is clear.
- The authors have provided code.
- The paper presents a number of benchmarks against with known RL algorithms.
- The paper also includes results for human players and against an agent exploring randomly. This is good to see.
- The procedural generated environment every episode to support learning of general policies is very nice.
- The gradual unlocking of agent capabilities/achievements as shown in Figure 4 is very good and something different from many RL environments.

# Weaknesses
- It is not clear what the actual novel contribution of the work is. Yes a new RL environment/domain is good but is this just a cut down Minecraft or is there something more compelling about this environment.

# Questions for the authors
- It is not clear to me what the goal of the game is. I'll admit not having played Minecraft so perhaps I'm missing something. Is it to unlock as many achievements possible? Or is it to maximise the score? What does a high score tell you about what you have achieved?
- With regards to the achievements; do these directly translate to the action space? In other words are the achievements the actions available to the agent at any given time? So this means we have a dynamic action space right? As the agent unlocks more achievements and hence the action space gets larger, presumably this will affect the learning curves/times as the exploration space becomes larger. It would be nice to see how the learning changes as the action space increases.

**Summary Of The Paper:**

This paper presents a new RL environment called Crafter that is somewhat inspired by the RL environments based around the Minecraft game. The paper argues that Minecraft is difficult to solve and that Crafter is a simpler 2D Minecraft like game that presents its own challenges for RL research.

Some of the key features are the ability to procedurally generate a new world for each episode (in order to learn general policies) and the ability for the agent to unlock new features/capabilities.

The authors have provided some benchmarks using known RL algorithms and has also provided results for human players.

**Summary Of The Review:**

The paper is well written and presents a new Minecraft inspired environment for RL research which has some novel features. However, the paper needs to do a better job explaining the research contribution.
Similarly any discussion of limitations or future work is lacking.

---

> ### Author Response · Authors · 2021-11-12
> **Response to Reviewer cMdw (Part 2)**
>
> > The paper is well written and presents a new Minecraft inspired environment for RL research which has some novel features. However, the paper needs to do a better job explaining the research contribution.
>
> Well-designed benchmarks have historically been a strong driver of research progress in deep learning and the ICLR 2022 Call for Papers mentions "software platforms" in its non-exhaustive list of relevant topics. We stress (1) the importance of evaluating multiple tasks within a single environment to reduce resource requirements and (2) the strong need for meaningful benchmarks for unsupervised RL. Additionally, we establish performance baselines for RL with and without rewards through 5 trained agents and a human baseline. Finally, we report insights into sophisticated emergent behaviors in the environment.
>
> > discussion of limitations or future work is lacking
>
> Thank you for this feedback! We will add a discussion of the following limitations and future work to the paper:
>
> - First, we tuned the difficulty of Crafter to be challenging yet not hopeless for current methods. As research progresses towards solving the challenges that are currently present, it may become useful to extend Crafter by new enemies, resources, items, and achievements. Being written purely in Python, Crafter can easily be extended in this way.
>
> - Moreover, grouping the 22 achievements into categories like "memory", "generalization", "exploration", etc would allow measuring agent abilities more abstractly. This could result in evaluations similar to bsuite, but would allow measuring all skills with a single training run and outside of symbolic toy envs. We did not include this in the paper because such a categorization is subjective and will become clearer as more researchers use the environment.

---

> ### Author Response · Authors · 2021-11-12
> **Response to Reviewer cMdw (Part 1)**
>
> Thank you for your detailed review! Below, we address your concerns by expanding on the relation between Crafter and Minecraft, explaining the goal in the environment and the aggregated score, clarifying the action space, the research contributions, and adding a discussion of limitations/future work.
>
> Please confirm whether our response fully addresses your concerns or whether there are any remaining concerns that motivate recommending rejection, which we will be more than happy to address.
>
> > Yes a new RL environment/domain is good but is this just a cut down Minecraft or is there something more compelling about this environment.
>
> Minecraft is an exciting environment for reinforcement learning research. However, training time is slow and many sophisticated long-horizon behaviors − that human players perform with ease − are completely out of reach for current RL methods. Crafter provides a testbed for learning such behaviors separately from the problem of 3D perception and navigation, resulting in much faster iteration times.
>
> In contrast to Minecraft, Crafter was optimized for research productivity from the ground up: it is easy to install with Python-only dependencies, is easy to modify, comes with 22 standardized tasks, a human performance baseline, code for 5 baseline RL agents, and allows full access to semantic information about the world. *The idea is not to replace Minecraft, but to accelerate the research progress towards solving it.*
>
> > It is not clear to me what the goal of the game is. I'll admit not having played Minecraft so perhaps I'm missing something. Is it to unlock as many achievements possible? Or is it to maximise the score?
>
> The goal is to unlock all 22 achievements in as many episodes as possible, as stated throughout the paper (including the abstract).
>
> As such, Crafter is a form of multi-task environment with 22 tasks that the agent can choose to solve in any feasible order. At the same time, Crafter benefits from the easy training protocol of standard RL. We see this as a fruitful − and resource effective − evaluation setting compared to many existing benchmark suites where each environment only measures one individual task.
>
> The aggregated score serves the same purpose as in existing benchmark suites. While Atari agents optimize the rewards of each individual game, they are evaluated by the human-normalized median score across games.
>
> > What does a high score tell you about what you have achieved?
>
> The score is the geometric mean over success rates for the 22 achievements. A score of 0% means the agent never unlocks any achievement and a score of 100% means the agent unlocks all achievements in all episodes. We recommend a geometric mean as aggregation because we prefer agents that unlock many achievements (including challenging ones) over agents that unlock easy  achievements very often. Thus, the geometric mean can be thought of as a weighted average that gives more weight to harder tasks.
>
> When developing the environment, we initially only measured the sum of rewards. However, we found that agents made vastly different progress in the game while showing similar rewards, because of the broad spectrum of abilities measured by Crafter. On the other hand, we did not want to guess how difficult each task might be and assign different rewards based on that, which might also differ between algorithms.
>
> The geometric mean is an elegant solution to this dilemma of aggregating scores of multiple tasks. It automatically weighs difficult tasks more strongly, without having to know a priori which tasks are difficult. We see this as a step towards better multi-task evaluation not just in Crafter but also in more complex environments that will be developed in the future. We will add the above discussion to Section 3.3.
>
> > With regards to the achievements; do these directly translate to the action space? In other words are the achievements the actions available to the agent at any given time? So this means we have a dynamic action space right?
>
> No, the action space is categorical and fixed. While unlocking the 22 achievements requires selecting the right actions in the right situations, there is no direct mapping between the two. For example, the "make_iron_pickaxe" has a corresponding action but the "defeat_zombie" and "wake_up" achievements don't.

---

> > ### Comment · Reviewer_cMdw · 2021-11-18
> > **Response to authors**
> >
> > Thank you for your replies and apologies for taking a while to get back to you.
> > You answers help to clarify some of the questions I had. It would be good if possible to incorporate some of these answers into the paper especially for readers who may not have familiarity with crafting type games.

---

> > > ### Author Response · Authors · 2021-11-18
> > > **Response to Reviewer cMdw**
> > >
> > > Thank you for acknowledging that we have clarified some of your questions. To improve the paper, we will elaborate on the relation to Minecraft, the goal of achieving the 22 tasks, and the interpretation of the score, based on our answers above. Do you have any remaining concerns about the paper that we should address? If so, it would be helpful to know the specific problems, so that we know what to improve. Thanks!

---

> > > ### Author Response · Authors · 2021-11-19
> > > **Paper updated**
> > >
> > > Dear reviewer,
> > >
> > > We have updated the paper to include the episode returns in Table 1, an interpretation of the score at the end of Section 3.3, and a discussion of future work in Section 5. The revision is now available on OpenReview.
> > >
> > > The paper already states that the agent is evaluated by 22 semantically meaningful achievements (shown in Fig 4) and that the action space is a flat categorical (Section 3.2 under "Actions"), but we would be happy to clarify further in a way you think would be helpful.
> > >
> > > With the discussion period coming to an end, we would highly appreciate if you could confirm whether these changes have resolved the main issues you raised and you are planning to update your recommendation, or whether there are any remaining specific issues.
> > >
> > > Thanks!

---

> > > ### Author Response · Authors · 2021-11-23
> > > **Reminder for Reviewer cMdw**
> > >
> > > Dear reviewer, we've updated the paper with your recommended changes a couple of days ago. Additionally, we pointed out that the paper already states the goal and the action space of the environment, but if you think it is insufficient, we would be happy to make it clearer based on a concrete suggestion from you. In either case, would you mind confirming the update and letting us know whether there are any remaining larger issues that we should address or increasing your score if the issues have been resolved? Thank you!

---

> ### Author Response · Authors · 2021-11-18
> **Reminder for Reviewer cMdw**
>
> Dear reviewer, with the discussion period ending soon, could you please confirm whether our response has fully addressed your concerns about your submission? If there are still any specific issues that prevent you from recommending acceptance, we will be more than happy to address them. Thanks!

---

### Official Review · Reviewer_Mm58 · 2021-11-02

**Correctness:** 4
**Technical Novelty And Significance:** 3
**Empirical Novelty And Significance:** Not applicable
**Recommendation:** 8
**Confidence:** 4

**Details Of Ethics Concerns:**

The acknowledgements were accidentally left in the submitted manuscript, partially violating the author anonymity policy.

**Main Review:**

### Originality

The environment proposed is quite similar to prior work (Mine RL): the game has very similar high level dynamics, with the main differences being that 1) Crafter is 2D rather than 3D, and that 2) Crafter is slightly simpler in terms of amounts of possible tasks (e.g. in Mine RL, the agent can craft many more things than in Crafter).

Additionally, it is similar to Nethack (which was recently proposed as an AI benchmark) in the 2D nature of the game and it's hierarchical long-horizon dynamics. However – as the authors mention – Nethack is much more challenging given the vast number of items that the agent needs to learn to recognize and use effectively. With Crafter, the authors aim to provide a benchmark that isn't as memorization heavy (even though they still claim that memorization is one of the main research challenges of Crafter in Sec. 3.4).

### Quality / Clarity

Overall, the paper was very well written and clear, and was a pleasure to read!

In 4.4 there is a typo: "harvest plans" -> "harvest plants"


### Limitations

In comparing to previous work, the authors state that "Minecraft is too complex to be solved by current methods (Milani et al., 2020), it is unclear by what metric agents should be evaluated by". It might be worth elaborating on the first part of the statement, as even in Crafter no agent is able to "solve" it (and in fact are very far behind humans, as mentioned in the manuscript). Regarding the second part of the statement, the lack of metrics is part of the appeal of the environment (see https://minerl.io/basalt/), so might be worth rephrasing.

One of the main selling points of Crafter is the relative sample-efficiency of training relative to other benchmarks. I thought that there was not enough space dedicated to analyzing the relative amount of training required for _convergence_ in Crafter relative to Atari, Mine RL, Nethack, and so on. While all the results for Crafter are capped at 1M timesteps, one could also do the same in these other environments. While I don't expect these agents would be very good at all, that would give a better sense of what order of magnitude of the difference between these environments is. Additionally, it would be nice to see (in the appendix) what the 50M timestep Dreamer-v2 training run looks like (especially, whether it has reached convergence at 50M).

Another thing that is curious to me is the reason behind evaluating the agents on the average success rate across the whole of training. I understand that this is to take into account sample-efficiency – but that is already being forced by capping the training to 1M timesteps. This additional choice of averaging across epochs seems to greatly penalize algorithms which might be very sample-inefficient at first, but then later catch up, leading to the same final performance as others. It would be nice to at least include the final-epoch success rates as additional information.

Another thing that is strange is that the random agent seems to achieve the same level of episode length as PPO agents. It would be useful to have some additional clarity about why this is.

**Summary Of The Paper:**

This paper introduces a new RL benchmark, Crafter, which is a simplified version of Minecraft. The aim of Crafter is to offer an environment which has many desirable properties: while it is a challenging environment in which complex behaviors are possible and necessary for the agents' survival, it's simultaneously simple enough to make training require much less interactions relative to prior benchmarks.

**Summary Of The Review:**

Overall, Crafter – while a straightforward simplification of another existing environment (Mine RL) – could become a useful benchmark task in the field. While it is not fully novel, it positions itself uniquely relative to other benchmarks across various axes relevant for research (pixel vs state based, memory vs not-memory intensive, 2D vs 3D complexity, small vs large required compute). The low amount of compute required for training (while preserving environment and potential behavior complexity) is worth emphasizing, as it could speed up the rate of progress of further research in this space.

It would however be nice to see addressed the limitations mentioned above.

---

> ### Author Response · Authors · 2021-11-12
> **Response to Reviewer Mm58**
>
> Thank you for your detailed review! Below, we respond to your concerns by elaborating that Crafter is a challenging benchmark but still easier than Minecraft, adding scores and code of the agent trained for 50M steps, including scores computed over the last 100 episodes, and providing clarity on the episode length achieved by PPO. If you have any remaining concerns, please let us know and we will be happy to address them.
>
> > In comparing to previous work, the authors state that "Minecraft is too complex to be solved by current methods (Milani et al., 2020), it is unclear by what metric agents should be evaluated by". It might be worth elaborating on the first part of the statement, as even in Crafter no agent is able to "solve" it (and in fact are very far behind humans, as mentioned in the manuscript). Regarding the second part of the statement, the lack of metrics is part of the appeal of the environment (see https://minerl.io/basalt/), so might be worth rephrasing.
>
> Thank you for the suggestions, we will expand on them in the paper. Regarding the first part, we agree that Crafter itself is far from unsolved. Our point is that we can make research progress on learning sophisticated long-term behaviors even in the absence of 3D worlds, resulting in much faster iteration times. The goal is not to replace Minecraft but to progress faster towards it!
>
> Regarding the second part, we agree and will change the wording. Specifically for RL with and without rewards, we believe that clear evaluation metrics are important for driving research progress. For those, Minecraft is appealing because of its open-endedness, with the agent being able to perform many different behaviors, but evaluation is nontrivial. Crafter comes with 22 tasks that are designed to broadly cover the meaningful skills in the environment. Together with its fast iteration times, this makes Crafter a productive RL benchmark, also for unsupervised agents that are not tied to specific tasks.
>
> > I thought that there was not enough space dedicated to analyzing the relative amount of training required for convergence in Crafter relative to Atari, Mine RL, Nethack, and so on. While all the results for Crafter are capped at 1M timesteps, one could also do the same in these other environments. While I don't expect these agents would be very good at all, that would give a better sense of what order of magnitude of the difference between these environments is.
>
> We agree that this would be a nice addition. While we see running several baseline agents on environments other than Crafter to be outside of the scope of this work, we will include the detailed performance metrics for the 50M DreamerV2 agent in the appendix, which achieved a reward of ~15, and also make its code available for reproducibility.
>
> > the reason behind evaluating the agents on the average success rate across the whole of training. I understand that this is to take into account sample-efficiency – but that is already being forced by capping the training to 1M timesteps. This additional choice of averaging across epochs seems to greatly penalize algorithms which might be very sample-inefficient at first, but then later catch up, leading to the same final performance as others. It would be nice to at least include the final-epoch success rates as additional information.
>
> Thank you for this suggestion. The main motivation for computing the scores across all training episodes is to measure "regret", which as you said puts an emphasis on sample efficiency. Moreover, computing success rates requires a certain number of episodes, at least 100 to compute percentages but ideally more for higher accuracy, especially to also compare agents by rarely completed tasks, such as finding a diamond. For references, we will also add success rates computed over the last 100 episodes.
>
> > the random agent seems to achieve the same level of episode length as PPO agents. It would be useful to have some additional clarity about why this is.
>
> Compared to the random agent, the PPO agent explores the map faster and thus is more likely to be exposed to monsters. From Appendix A, we can see that PPO has not yet learned to reliably defeat zombies and skeletons, compared to Rainbow and DreamerV2. Despite this, PPO manages to unlock some of the harder tasks, such as crafting tools, collecting coal, and placing stones and furnaces, resulting in an overall higher score than Rainbow.

---

### Decision · Program_Chairs · 2022-01-20

**Decision:**

Accept (Poster)

**Comment:**

This paper introduces a new RL benchmark that is a simplified 2D version of Minecraft -- it is designed to support complex behaviors but reduce the training complexity. It is very well written and clear, positioned well with respect to other benchmarks, and is likely to improve the speed of development/testing of some RL algorithms. It is likely to appeal to a subset of the community and drive research in some cases, while others may prefer to stick with full 3D Minecraft. As such, there are some mixed reviews on the paper, with open questions as to whether it would be welcomed by people who work on Minecraft-style domains, whether behaviors learned in the simplified 2D environment would generalize to other settings/domains, and the potential for agents to game the environment. The authors are encouraged to take these aspects and perspectives into consideration when revising the paper.